# Induced dipole moments in amorphous ZnCdS catalysts facilitate photocatalytic H₂ evolution

Xin Wang[1], Boyan Liu[1], Siqing Ma[1], Yingjuan Zhang[1], Lianzhou Wang [2] ✉, Gangqiang Zhu [3] ✉, Wei Huang [1] ✉ & Songcan Wang [1] ✉

Amorphous semiconductors without perfect crystalline lattice structures are usually considered to be unfavorable for photocatalysis due to the presence of enriched trap states and defects. Here we demonstrate that breaking long-range atomic order in an amorphous ZnCdS photocatalyst can induce dipole moments and generate strong electric fields within the particles which facilitates charge separation and transfer. Loading 1 wt.% of low-cost Co-MoSₓ cocatalysts to the ZnCdS material increases the H₂ evolution rate to 70.13 mmol g⁻¹ h⁻¹, which is over 5 times higher than its crystalline counterpart and is stable over the long-term up to 160 h. A flexible 20 cm × 20 cm Co-MoSₓ/ZnCdS film is prepared by a facile blade-coating technique and can generate numerous observable H₂ bubbles under natural sunlight, exhibiting potential for scale-up solar H₂ production.

Carbon neutrality has been the global consensus for sustainable development in the modern society[1]. Owing to the high energy capacity (143 MJ kg⁻¹) and carbon-free features, hydrogen (H₂) has been regarded as a promising energy source to replace fossil fuels in the future[2]. However, over 95% of H₂ in the world is produced from fossil fuels such as steam reforming of natural gas, partial oxidation of heavier hydrocarbons, and coal gasification, which not only consume nonrenewable fossil fuels, but also generate carbon dioxide[3]. Therefore, seeking efficient, low-cost and green H₂ production technology is of paramount importance.

Photocatalytic water splitting using solar energy provides a cost-effective and environmental-friendly way for green H₂ production[4–6]. Nevertheless, the large-scale application of this technology is mainly restricted by its low H₂ production efficiency and poor stability, due to severe charge recombination in the bulk and strong redox capacities of the photogenerated electron-hole pairs that may decompose the photocatalyst itself[7,8]. Therefore, the development of highly-active, robust and cost-effective photocatalysts is significant while challenging. If the separation and transfer properties of the photogenerated electron-hole pairs can be significantly improved, the photogenerated electrons and holes can be consumed in the surface photocatalytic reactions, and thus side-reactions between the photogenerated charge carriers and the photocatalyst itself can be eliminated. During a photocatalytic process, the photogenerated electrons and holes are transported randomly within a particulate photocatalyst due to the lack of directed electric field[9]. Consequently, only a small proportion of photogenerated charge carriers can reach the surfaces of the photocatalyst for water splitting reactions[10]. In the past decades, worldwide efforts have been devoted to promoting charge separation by manipulating the built-in electric field in photocatalysts[11–13]. For example, a built-in electric field in Ag₂O-BaTiO₃ was generated by polarization of ferroelectric BaTiO₃[14], which generally needs external mechanical energy (e.g. ultrasonication). A built-in electric field also can be formed by heterojunction in photocatalysts, but only a different space charge region is generated at the interface of two materials rather than a desirable bulk band bending[15–17]. As a result, the bulk charge recombination issue cannot be completely addressed. How to substantially enhance the bulk

[1]Frontiers Science Center for Flexible Electronics, Xi'an Institute of Flexible Electronics (IFE), Northwestern Polytechnical University, 127 West Youyi Road, Xi'an 710072, China. [2]Nanomaterials Centre, Australian Institute for Bioengineering and Nanotechnology and School of Chemical Engineering, The University of Queensland, Brisbane, QLD 4072, Australia. [3]School of Physics and Information Technology, Shaanxi Normal University, Xi'an 710062, China. ✉e-mail: l.wang@uq.edu.au; zgq2006@snnu.edu.cn; iamwhuang@nwpu.edu.cn; iamscwang@nwpu.edu.cn

charge separation efficiency in particulate photocatalysis remains an open question.

If the symmetry of the atom arrangement in the crystal structure of a photocatalyst is broken, the positive and negative charge center will shift to form a dipole moment and generate an electric field[18–21], which may provide alternative opportunities to address the bulk charge recombination issue during photocatalysis. Here, we develop a facile wet-chemical synthesis process for the design of amorphous ZnCdS (AZCS) photocatalyst. Both density functional theory (DFT) simulation and experimental characterizations demonstrate that the asymmetric atom arrangement in AZCS can generate dipole moments to form electric fields which facilitates charge separation and transfer. By loading Co-MoS$_x$ as a low-cost cocatalyst, the Co-MoS$_x$/AZCS can produce numerous H$_2$ bubbles under both Xe lamp light and natural sunlight (Supplementary Movies 1 and 2), exhibiting a photocatalytic H$_2$ evolution rate of 70.13 mmol g$^{-1}$ h$^{-1}$, which is over 5 times higher than its crystalline counterpart (13.90 mmol g$^{-1}$ h$^{-1}$). Furthermore, an apparent quantum yield (AQY) of 38.54% is obtained from Co-MoS$_x$/AZCS illuminated by a 420 nm monochromatic light. The Co-MoS$_x$/AZCS demonstrates a long-term stability of up to 160 h for photocatalytic H$_2$ evolution. In addition, a flexible Co-MoS$_x$/AZCS film on an aluminum foil is fabricated by a blade-coating approach, which is robust after repeated bending (Supplementary Movie 3), and a large number of H$_2$ bubbles can be observed under both Xe lamp light and natural sunlight irradiation (Supplementary Movies 4 and 5).

## Results

### Mechanism of amorphous structure induced dipole fields

To theoretically verify that amorphous ZnCdS (AZCS) has stronger dipole fields due to the disorder arrangement of atoms, density functional theory (DFT) calculation was carried out to investigate the effect of amorphous structure on the separation and transport of photogenerated carriers. In the hexagonal system of crystalline ZnCdS (CZCS), each Zn or Cd atom is connected to four S atoms with a perfect layered structure (Fig. 1a and top view in Supplementary Fig. S1a), while AZCS exhibits a random arrangement of the ZnS$_4$ and CdS$_4$ tetrahedrons (Fig. 1b and top view in Supplementary Fig. S1b). Supplementary Fig. S2 exhibits the DFT energy as a function of time at 300 K for CZCS and AZCS, respectively. CZCS exhibits a lower energy than its

AZCS counterpart, demonstrating the higher structural stability. Moreover, the deformation charge density distributions of CZCS and AZCS along the (011) plane are shown in Fig. 1c, d, respectively. The charge distribution of CZCS is very uniform and highly order, while AZCS demonstrates the random distribution of the deformation charge density, which is attributed to the different atomic arrangement and distribution in CZCS and AZCS.

According to the structures of AZCS and CZCS, the dipole moments were calculated. As shown in Fig. 1e, CZCS with a perfect crystalline structure exhibits relatively small dipole moments of 31.73, −27.48, and −87.85 eÅ along the $x$-, $y$-, and $z$-directions, respectively. However, the dipole moment along the $x$-direction is significantly enhanced to −80.58 eÅ in AZCS. More obvious enhancement is observed in the $y$- and $z$-directions, with the values of −153.80 and 197.01 eÅ, respectively. When a pair of opposite charges "+$q$" and "−$q$" are separated by a distance "$d$", an electric dipole is established. The size of dipole is measured by its dipole moment, which is equal to $d$ multiplied by $q$. The direction of the dipole moment in space is from the negative charge "−$q$" to the positive one "+$q$"[22,23]. The larger of the absolute value of the dipole moment means the stronger of the extra driving force can be generated in a photocatalyst to promote charge separation. Therefore, the amorphous structure with disorder arrangement of atoms in AZCS induces strong dipole fields along the (100), (010) and (001) directions, thus facilitating charge separation and transfer in a particulate photocatalyst.

According to the calculation results shown in Fig. 1e, the distributions of the positive and negative charge centers of CZCS and AZCS in the $y$-$z$ plane are demonstrated (Fig. 1f, g). Since the distribution of charge density in CZCS is symmetrical (Fig. 1c), the positive and negative charge centers are close, thus generating a relatively small dipole moment (Fig. 1f). In comparison, the AZCS counterpart is completely asymmetrical (Fig. 1d), and the positive and negative charge centers are significantly separated, thus forming a much stronger dipole moment of 197.01 eÅ along the (001) direction (Fig. 1g).

To illustrate the contribution of dipole moments to charge separation, a schematic (Fig. 1h) of the energy band structures of CZCS and AZCS during the photocatalytic process were presented. For CZCS without obvious dipole moments, the energy band bending is too

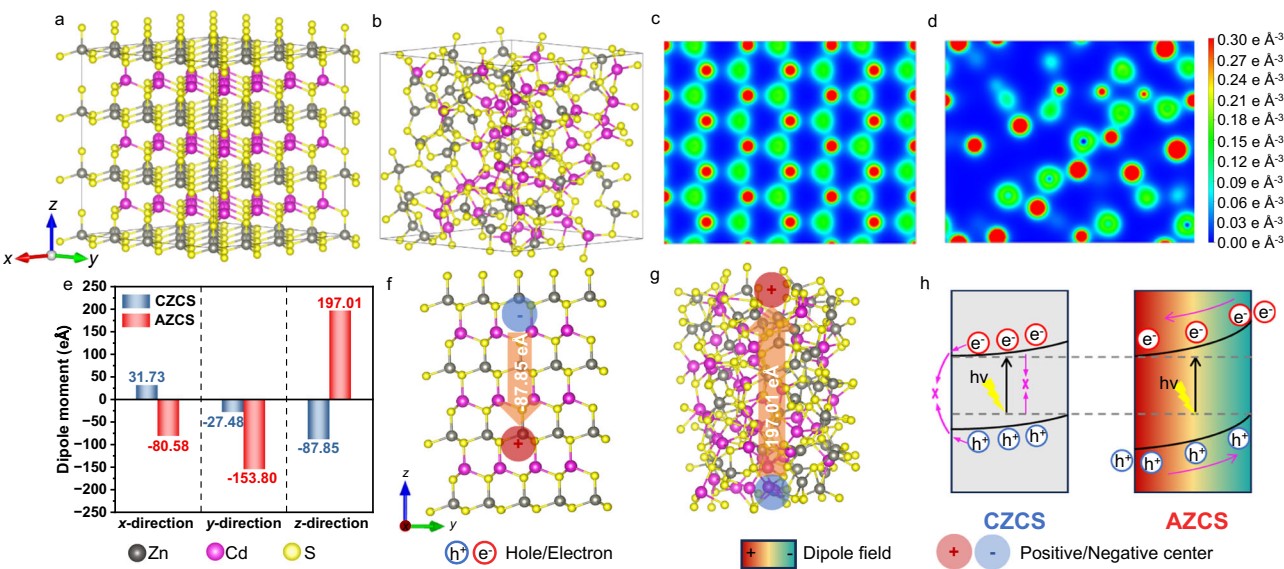

**Fig. 1 | Theoretical analysis of dipole fields in crystalline and amorphous structures.** The atom arrangement and distribution of (**a**) CZCS and (**b**) AZCS. Deformation charge densities of (**c**) CZCS and (**d**) AZCS on the (011) plane. **e** The calculated dipole moments of AZCS and CZCS along three different crystallographic directions. Schematics of (**f**) CZCS and (**g**) AZCS structures with positive and negative charge centers. **h** Schematic of the promotion effect of dipole field on charge transfer.

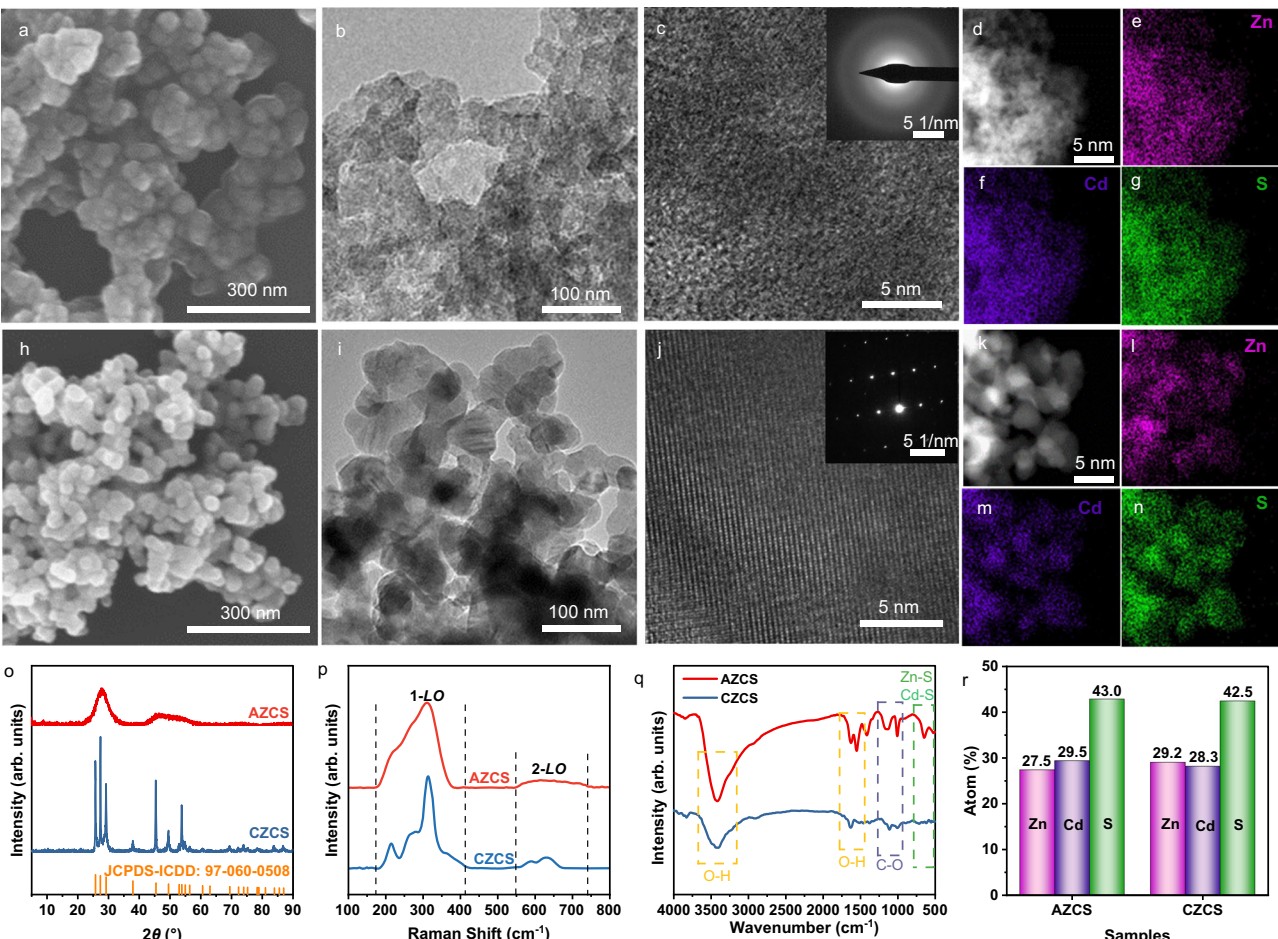

**Fig. 2 | Morphology, structure and surface characteristics of AZCS and CZCS.**
**a–c** SEM, TEM, HRTEM images and SAED pattern (inset in **c**) of AZCS, (**d–g**) TEM EDS and elemental mapping of Zn, Cd and S in AZCS, (**h–j**) SEM, TEM, HRTEM images and SAED pattern (inset in **j**) of CZCS, (**k–n**) TEM EDS and elemental mapping of Zn, Cd and S in CZCS, (**o–q**) XRD patterns, Raman and FTIR spectra of AZCS and CZCS, (**r**) The relative element content of Zn, Cd, S from AZCS and CZCS obtained by ICP-OES.

small to drive the directional separation of the photogenerated electrons and holes, and severe charge recombination occurs. However, when the order of all well-arranged unit cells in the ZnCdS crystal are disrupted, it will cause uneven charge distribution in space, thereby generating dipole moments that induce strong dipole fields in the entire photocatalyst. The strong directional dipole field will cause a large energy band bending in AZCS to promote charge separation, which can enhance the photocatalytic activity and stability.

To demonstrate the effect of atomic arrangement and distribution on the bandgap, the highest occupied molecular orbital (HOMO) and least unoccupied molecular orbital (LUMO) of CZCS and AZCS were also calculated. As shown in Supplementary Fig. S3, the HOMO charge densities are strongly localized at the S atoms, and the LUMO charge densities are strongly localized at the Zn, Cd, and S atoms, which is consistent to the literature that the valence band (VB) maximum of ZnCdS is mainly dominated by the 3$p$ orbital of the S atom, while the conduction band (CB) minimum of ZnCdS is mainly contributed by the hybridization of the 4$s$ orbital of the Zn atom, the 5$s$ orbital of the Cd atom and the 3$p$ orbital of the S atom[24]. In addition, the bandgap of AZCS is narrower than its CZCS counterpart, suggesting that AZCS can absorb a broader range of light.

## Characterizations of the as-synthesized photocatalysts
The morphology and microstructure of the as-synthesized AZCS and CZCS were analyzed by scanning electron microscopy (SEM), transmission electron microscopy (TEM) and high-resolution TEM

(HRTEM). As shown in Fig. 2a, AZCS is composed of connected nanoparticles with a size of approximately 42.4 nm. No sharp grain boundaries can be observed in the TEM image (Fig. 2b), indicating the amorphous characteristics. The HRTEM image of AZCS shows disordered atomic arrangement without obvious lattice fringes (Fig. 2c), and the selected area electron diffraction (SAED) pattern exhibits diffused continuous and thick halo rings without any distinguishable diffraction spots (inset in Fig. 2c), indicating the amorphous feature of AZCS. Energy dispersive X-ray spectrometry (EDS) mapping of AZCS demonstrates the homogeneous distribution of Zn, Cd, and S elements (Fig. 2d–g). In comparison, numerous nanoparticles with a size of around 49.1 nm is observed in CZCS, and obvious grain boundaries between the particles can be observed (Fig. 2h, i), which are attributed to the increased crystallinity during high temperature sintering.

According to the SEM and TEM images of AZCS and CZCS (Fig. 2a, h), their particle sizes are similar. To further confirm the particle size distributions, the AZCS and CZCS powders were characterized by a nanoparticle size analyzer. As demonstrated in Supplementary Fig. S4, the particle size distribution of AZCS is 100–250 nm, while that of CZCS is 100–350 nm. The average particle sizes of AZCS and CZCS are 170.47 and 213.43 nm, respectively, suggesting that high temperature sintering at 600 °C enlarges the particle size distribution range while has little effects on the average particle size. It should be mentioned that the particle size measured by a nanoparticle size analyzer is the statistical results from 20 mg of the sample, while the particle size measured by TEM is only the observable

particles shown in the TEM image. Therefore, the particle size values measured by these two different methods may be different, while the values measured by the same method can be reasonably compared. The ordered atomic arrangement is reflected by the clear lattice fringes in the HRTEM image (Fig. 2j), and the matrix spots in the SAED pattern (inset in Fig. 2j) demonstrate that CZCS is highly crystalline. EDS mapping confirms the homogeneous distribution of Zn, Cd and S (Fig. 2k–m), which is similar to that of its AZCS counterpart.

The crystal structures of AZCS and CZCS were further revealed by X-ray diffraction (XRD). As shown in Fig. 2o, distinctively different XRD patterns are observed between AZCS and CZCS. Specifically, CZCS exhibits extremely sharp diffraction peaks while AZCS has only two broad peaks, which proves the long-range disorder of atomic arrangement in AZCS. In addition, the XRD pattern of CZCS matches well with hexagonal ZnCdS (JCPDS-ICDD: 97-060-0508).

To understand the effect of temperature on the crystallinity during synthesis and sintering, other AZCS samples were prepared at room temperature, followed by sintering at different temperatures (200, 400, and 600 °C). Compared to the XRD pattern of ZCS0, the wide peak at 40–60° is divided into two peaks when the sample is prepared at room temperature (ZCSRT, Supplementary Fig. S5), indicating the increase of crystallinity. In addition, with the increase of temperature during sintering, all peaks in the XRD patterns become sharper (Supplementary Fig. S5), suggesting the gradual increase of crystallinity. The relative crystallinity of the ZCS0, ZCSRT, ZCS200, ZCS400, and ZCS600 samples was calculated according to their XRD peaks. As listed in Supplementary Table S1, the crystallinity of ZCS0 (AZCS) is only $16.20 \pm 1.62\%$ while its ZCS600 (CZCS) counterpart is $90.52 \pm 1.25\%$. The crystallinities of ZCSRT, ZCS200 and ZCS400 are $32.43 \pm 1.39\%$, $49.37 \pm 2.26\%$ and $65.68 \pm 2.90\%$, respectively, which is in line with the rule that the crystallinity increases with the increase of the sintering temperature.

To better demonstrate the evolution of XRD peaks during sintering, in situ XRD characterization was carried out for a ZCS0 sample sintering from room temperature to 600 °C in a $N_2$ atmosphere. Supplementary Fig. S6 shows similar trends to the ex-situ results (Supplementary Fig. S5), confirming the evolution of amorphous to crystalline features when the temperature is increased from room temperature to 600 °C. It should be mentioned that the XRD peaks of the sample collected at 600 °C by in-situ XRD (Supplementary Fig. S6) are weaker than the ZCS600 characterized by ex-situ XRD (Supplementary Fig. S5), which is attributed to the much higher scanning rate during in situ characterization (2°/min for ex situ XRD, while 5°/min for in situ XRD).

Raman and Fourier transform infrared (FTIR) spectra were carried out to further characterize the surface chemical bonding of AZCS and CZCS. As shown in Fig. 2p, two characteristic Raman bands at $173.52–412.41$ cm$^{-1}$ and $548.70–714.67$ cm$^{-1}$ are attributed to the 1st and 2nd longitudinal-optical (1-LO and 2-LO) phonons in ZnCdS, respectively[25]. AZCS exhibits two broader and less distinct bands at 311 and 620 cm$^{-1}$ compared to its CZCS counterpart, demonstrating the spatial disorder and translational asymmetry of the amorphous structure[26]. The FTIR spectra of AZCS and CZCS are shown in Fig. 2q. The FTIR vibrational peaks lie in the ranges of $3200–3500$ and $1580–1630$ cm$^{-1}$, corresponding to the O-H group stretching and bending vibrations respectively, which represents the adsorption of $H_2O$ molecules on the samples. In addition, the stretching vibration bands of Zn-S and Cd-S are observed in the range from 500 to 750 cm$^{-1}$. The other distinct band at $1000–1200$ cm$^{-1}$ should be ascribed to the C-O-C bonds, which is due to the surface adsorption of $CO_2$ from the air[27]. It is obvious that the band intensity of all bonds in CZCS are weaker than its AZCS counterpart, indicating the adsorption of less $H_2O$ and $CO_2$ molecules, possibly due to the decrease of surface energy in the crystalline structure[28].

Since sulfur tends to escape from the crystal structure of metal sulfides during high temperature sintering, inductively coupled plasma (ICP) was performed to investigate the possible change of element content before and after crystallization. As illustrated in Fig. 2r, the atomic ratios of Zn, Cd, S in AZCS and CZCS are almost unchanged, suggesting that thermal treatment at 600 °C only significantly increases the crystallinity while has little effects on the relative elemental content of the samples. Thermogravimetric (TG) analysis also confirms that AZCS is relatively stable at 600 °C (Supplementary Fig. S7, Supplementary Discussion). X-ray photoelectron spectroscopy (XPS) demonstrate that the valence states of all elements in AZCS and CZCS are almost unchanged (Supplementary Fig. S8, Supplementary Discussion).

## Photocatalytic $H_2$ evolution performance

Photocatalytic $H_2$ evolution performances of AZCS and CZCS were evaluated upon the irradiation of a Xe lamp using 20 mg of photocatalyst loaded with 1 wt.% of Co-MoS$_x$ as the cocatalyst in water containing lactic acid as a hole sacrificial agent. As shown in Fig. 3a, CZCS demonstrates a $H_2$ evolution rate of about 13.90 mmol g$^{-1}$ h$^{-1}$. In comparison, the $H_2$ evolution rate of AZCS reaches 70.13 mmol g$^{-1}$ h$^{-1}$, which is over 5 times higher than its CZCS counterpart. Supplementary Movies 1 and 2 present the photocatalytic $H_2$ evolution performance of AZCS under the irradiation of a Xe lamp and natural sunlight, respectively. A large number of $H_2$ bubbles can be observed, further indicating the superior $H_2$ evolution activity of AZCS. To understand whether the interfacial binding of the photocatalyst and the cocatalyst will affect the photocatalytic activity, Raman spectra (Supplementary Fig. S9) of Co-MoS$_x$ loaded on AZCS and CZCS, and the photocatalytic $H_2$ evolution performances of AZCS and CZCS loaded with different cocatalysts (Supplementary Figs. S10–12) were performed. The results prove that the cocatalyst does not affect the relative activity of AZCS and CZCS, while playing a pivotal role in accelerating surface photocatalytic reactions to alleviate the side reactions between the photocatalyst and the photogenerated charge carriers (Supplementary Discussion).

To study the photocatalytic activity of ZnCdS with different crystallinities, photocatalytic $H_2$ evolution performances of ZnCdS synthesized at 0 °C, RT, 200 °C, 400 °C and 600 °C (denoted as ZCSx, x is the synthesis temperature) were also measured. As shown in Supplementary Fig. S13, with the increase of the preparation temperature, the photocatalytic activity of ZCSx is gradually decreased (the $H_2$ evolution rates of ZCS0, ZCSRT, ZCS200, ZCS400 and ZCS600 are 70.13, 45.12, 33.47, 28.69 and 13.90 mmol g$^{-1}$ h$^{-1}$, respectively), which illustrates that the crystallinity is inversely proportional to the photocatalytic activity. Besides, the photocatalytic $H_2$ evolution rates of ZnCdS with different molar mass ratios of Zn/Cd are shown in Supplementary Fig. S14a. The $H_2$ evolution activity is distinctively improved with the decrease of the molar mass ratio of Zn/Cd. Until the molar mass ratio of Zn/Cd reaches 1:1, the $H_2$ evolution rate achieves the highest of 70.13 mmol g$^{-1}$ h$^{-1}$. When the molar mass ratio of Zn/Cd further decreases, the $H_2$ evolution rate also decreases. XRD patterns of the samples with different Zn/Cd ratios are shown in Supplementary Fig. S14b. With the increase of the Cd content, the peak at around 25° shifts to higher $2\theta$ degree, which is due to the lattice distortion caused by the incorporation of Cd with a larger atomic radius[29]. Three diffraction peaks of ZnS, Zn$_{4/5}$Cd$_{1/5}$S and Zn$_{3/5}$Cd$_{2/5}$S transform into two peaks in Zn$_{1/2}$Cd$_{1/2}$S, Zn$_{2/5}$Cd$_{3/5}$S and Zn$_{1/5}$Cd$_{4/5}$S, indicating that the addition of appropriate amount of cadmium in the low-temperature synthesis process can affect its crystallinity[30].

Figure 3b and Supplementary Fig. S15 present the apparent quantum yield (AQY) values of AZCS and CZCS at different monochromatic light and the detailed results are listed in Supplementary Table S2. The wavelength-dependent AQY variation trend is almost consistent with the absorption spectrum in both AZCS and CZCS,

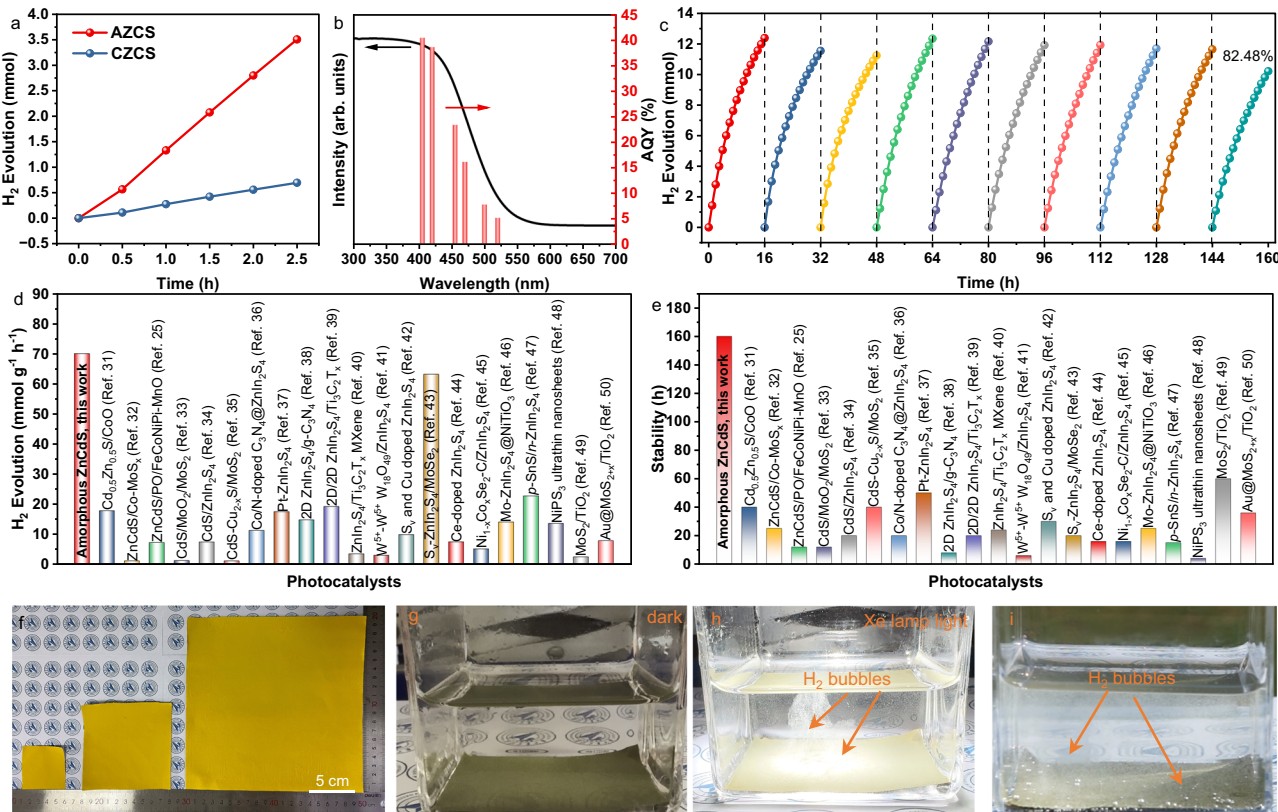

**Fig. 3 | Photocatalytic H₂ evolution performance. a** H₂ evolution within 2.5 h for AZCS and CZCS, (**b**) Wavelength-dependent apparent quantum yield (AQY) and (**c**) photocatalytic stability test of AZCS, (**d**) H₂ evolution rate and (**e**) stability of different sulfide photocatalysts reported so far, (**f**) Flexible photocatalyst films of different sizes (5/10/20 cm × 5/10/20 cm), the H₂ evolution state of the photocatalyst film in (**g**) darkness, (**h**) Xe lamp light and (**i**) natural light conditions.

suggesting the same photocatalytic mechanism. Specifically, the AQY values of AZCS are 40.35% (405 nm), 38.54% (420 nm), 23.28% (455 nm), 16.01% (470 nm), 7.65% (500 nm), and 5.03% (520 nm), respectively. It can be observed that the AQY values of AZCS are much higher than that of CZCS under different monochromatic light, indicating the favorable light absorption and utilization efficiency of AZCS.

In addition to the obvious photocatalytic water splitting activity, stability is another pivotal factor for the practical application of photocatalysis. As shown in Fig. 3c, the AZCS/Co-MoS$_x$ sample shows excellent stability without obvious decay in the H₂ evolution performance after 160 h of repeated testing (16 h per cycle for 10 cycles with an additional 0.5 wt.% Co-MoS$_x$ cocatalyst added in each cycle). It is worth noting that when no additional Co-MoS$_x$ cocatalyst was added during each cycle, the H₂ evolution performance dropped rapidly after 5 cycles (Supplementary Fig. S16). In order to exclude the promotion effect of the amount of cocatalyst on the photocatalytic performance, the effect of different amounts of Co-MoS$_x$ on the H₂ evolution performance of AZCS was investigated. As shown in Supplementary Fig. S17, the H₂ evolution performance of AZCS is the best when the Co-MoS$_x$ content is 1 wt.%. When the Co-MoS$_x$ cocatalyst content is further increased to 1.5 and 2 wt.%, the obtained AZCS/Co-MoS$_x$ photocatalysts exhibit very similar photocatalytic performance compared to their counterpart with 1 wt.% of Co-MoS$_x$ cocatalyst. SEM, XRD and ICP were used to characterize the mechanism of Co-MoS$_x$ in promoting the photocatalytic stability of AZCS, and the results demonstrate that the morphology, structure, and elements in AZCS are relatively stable after photocatalytic H₂ evolution test (Supplementary Figs. S18, 19 and Supplementary Table S3, Supplementary Discussion).

The photocatalytic H₂ evolution rates and stability of typical sulfide photocatalysts in recent years are summarized in Fig. 3d, e. Among

various sulfide photocatalysts[25,31–50], the amorphous ZnCdS/Co-MoS$_x$ in this work exhibits the highest photocatalytic H₂ evolution rate (Fig. 3d). Moreover, metal sulfide semiconductors are traditionally considered to be less stable due to the easy oxidation of S$^{2-}$ during photocatalysis[51]. This work demonstrates that the addition of Co-MoS$_x$ cocatalyst during each photocatalytic cycle can achieve a superior stability as high as 160 h, which is much higher than other sulfide photocatalysts (Fig. 3e). In addition, Supplementary Figs. S20a, b exhibit the comparison of AQY and H₂ evolution rate of the reported photocatalysts with built-in electric fields or amorphous structures, as well as sulfide and COF-based photocatalysts in recent years. The AQY in this work holds a competitive advantage (Supplementary Fig. S20a), indicating promising development prospects for AZCS/Co-MoS$_x$. In addition, compared with other photocatalysts, the amorphous ZnCdS demonstrated in our work exhibits a H₂ evolution rate as high as 70.13 mmol g$^{-1}$ h$^{-1}$, which outperforms all these photocatalysts without the formation of heterojunctions or the addition of piezoelectric materials (Supplementary Fig. S20b).

For possible scale-up applications, particulate photocatalysis also meet the issues in terms of the separation and recycle of the photocatalysts. To address these issues, we demonstrated that flexible Co-MoS$_x$/AZCS films with dimensions of 5 cm × 5 cm, 10 cm × 10 cm, and 20 cm × 20 cm can be prepared by a facile blade-coating technique on aluminum foils (Fig. 3f). Owing to the flexible feature of the aluminum foil, the obtained Co-MoS$_x$/AZCS films are flexible and keep stable after repeated bending (Supplementary Movie 3). Impressively, the Co-MoS$_x$/AZCS films can generate observable H₂ bubbles under both Xe lamp light (Fig. 3g, h, Supplementary Movie 4) and natural sunlight irradiation (Fig. 3i, Supplementary Movie 5), indicating the potential for possible scale-up solar-driven H₂ production.

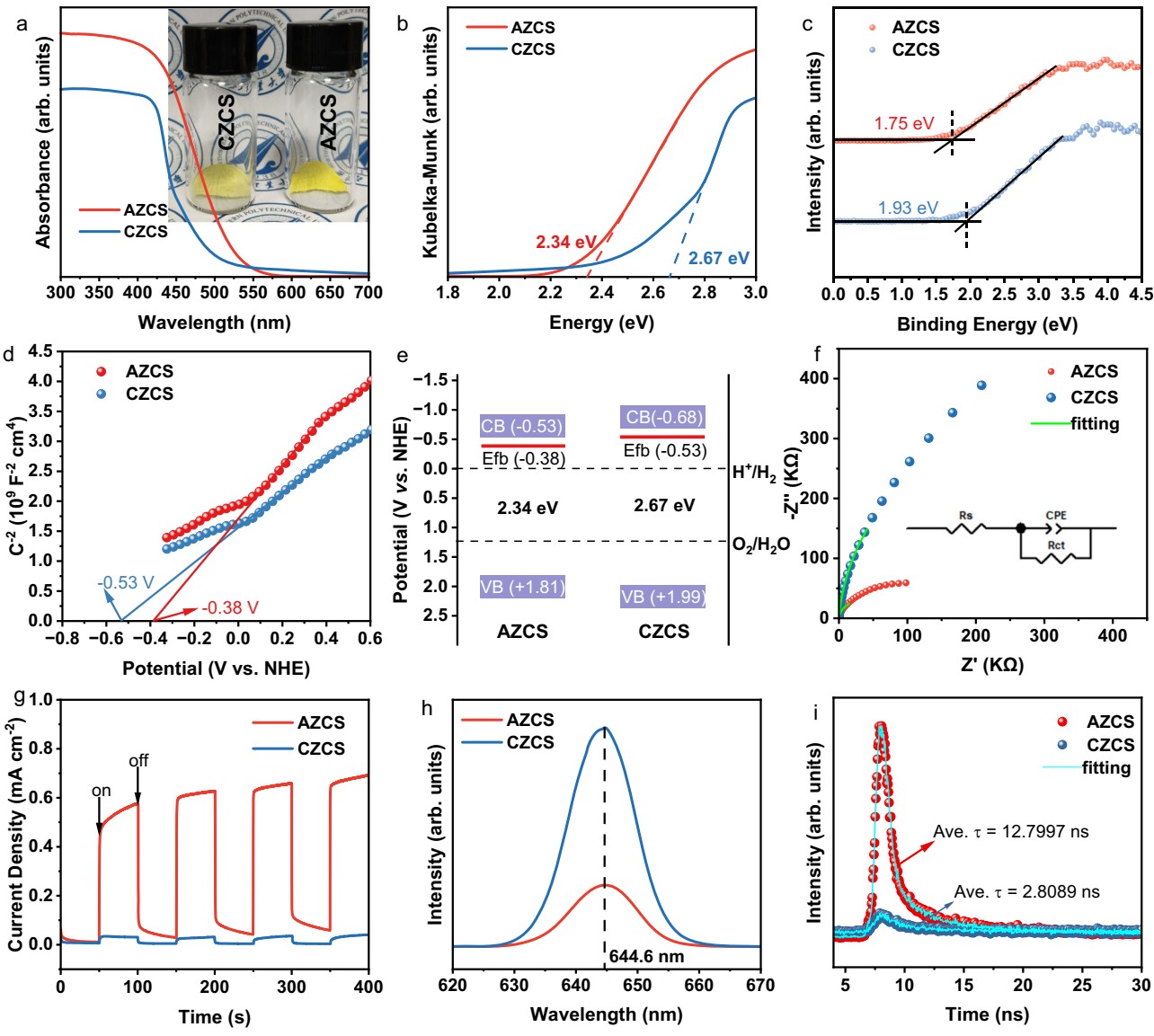

**Fig. 4 | Photocatalytic mechanism analysis. a** UV-vis light absorption curves of AZCS and CZCS, where inset shows the photograph of AZCS and CZCS, (**b**) the corresponding Tauc plots for AZCS and CZCS transformed by Kubelka-Munk parameter, (**c**) VB-XPS spectra, (**d**) M-S curves, (**e**) Band structures, (**f**) EIS plots (inset: equivalent circuit), (**g**) Transient photocurrent responses, (**h**) Photoluminescence (PL) spectra and (**i**) PL lifetime of AZCS and CZCS.

## Photocatalytic mechanism exploration

The successful progress of photocatalytic reactions requires photocatalysts to satisfy both thermodynamic and kinetic conditions. UV-vis absorption spectra, VB-XPS spectra and Mott-Schottky tests were conducted to understand the mechanism of the enhanced photocatalytic performance of AZCS. Figure 4a is the UV-vis absorption spectra of AZCS and CZCS. It is apparent that the absorption intensity of AZCS is higher than CZCS, indicating the better utilization of the solar spectrum. In addition, AZCS shows a red shift of the absorption edge, indicating the absorption of a broader range of light. Interestingly, AZCS exhibits a bright yellow color while CZCS shows a dark yellow color, indicating the absorption of different range of light (inset in Fig. 4a). Based on the UV-vis absorption spectra, the bandgap values ($E_g$) of AZCS and CZCS were obtained by the Kubelka-Munk function versus the light energy[52]. As demonstrated in Fig. 4b, AZCS presents a narrow bandgap of 2.34 eV, which is 0.33 eV lower than that of CZCS (2.67 eV).

To understand the arrangement of the band structures, VB-XPS spectra of AZCS and CZCS were collected, as shown in Fig. 4c.

The VB maximum values of AZCS and CZCS were calculated to be 1.81 and 1.99 eV using the formula of $E_{NHE}/V = \Phi + VB - 4.44$ ($E_{NHE}$: potential of normal hydrogen electrode, $\Phi$ is 4.5 eV representing the electron work function of the analyzer) to eliminate the influence of contact potential difference between the analyzer and the samples[53]. According to the equation of $E_{VB} = E_{CB} + E_g$ ($E_{CB}$ is the potential of CB)[54], and the $E_{CB}$ of AZCS and CZCS can be estimated to be −0.53 and −0.68 eV, respectively. The flat-band potentials ($E_{fb}$) of AZCS and CZCS are 0.38 and 0.53 eV, respectively, which are calculated by the Mott-Schottky plots in Fig. 4d. The plots of AZCS and CZCS both show a positive slope, which is typical for n-type semiconductors. Based on the above calculation and analysis, the band positions of AZCS and CZCS are demonstrated in Fig. 4e. Notably, the band alignments of AZCS and CZCS both satisfy the thermodynamic requirements for water reduction and oxidation, while AZCS exhibits a narrower bandgap due to the localization of band tail states in the amorphous structure[55]. The reduced bandgap of AZCS is consistent to the DFT calculation results shown in Supplementary Fig. S3.

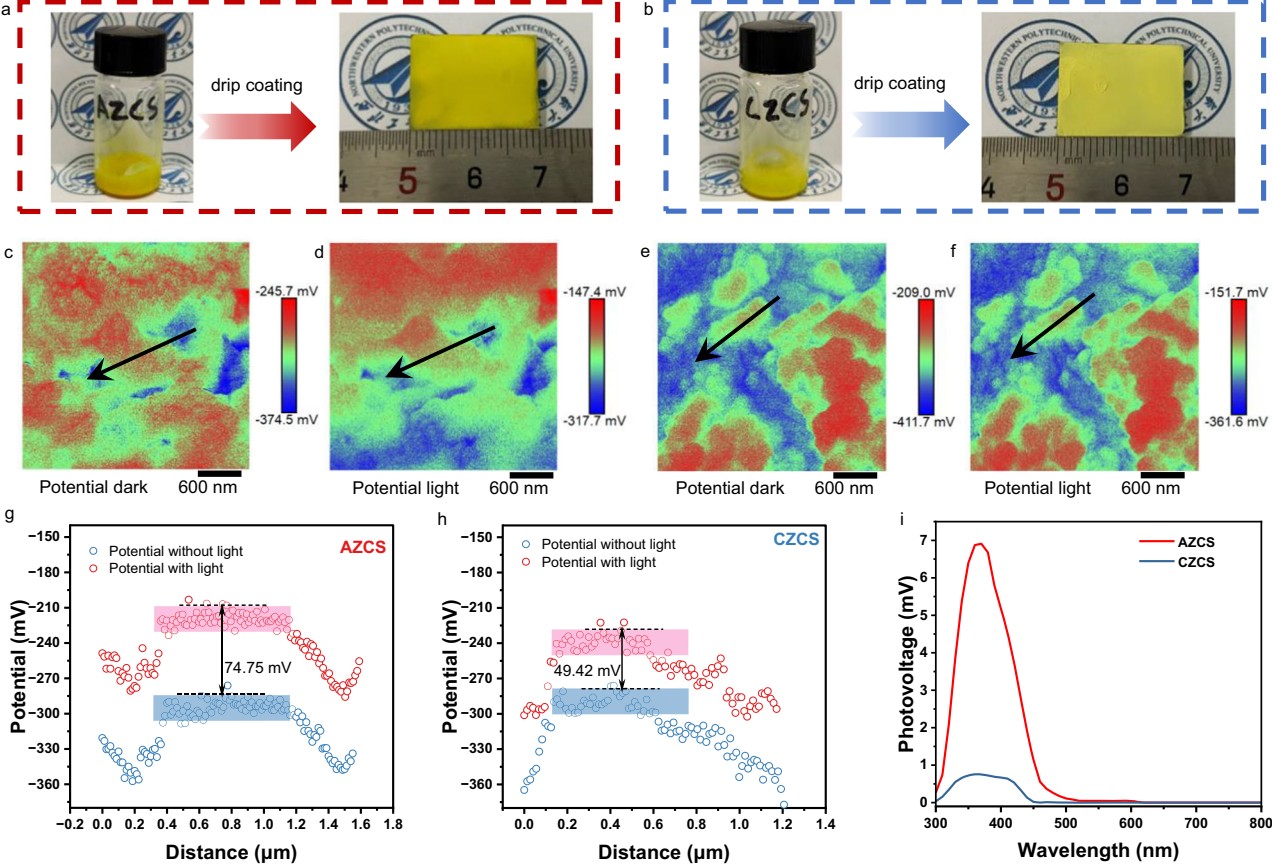

**Fig. 5 | Dipole field characterizations.** Digital images showing the slurries and drip-coated films of (**a**) AZCS and (**b**) CZCS. KPFM images of AZCS in (**c**) dark and (**d**) light. KPFM images of CZCS in (**e**) dark and (**f**) light. **g** Surface potential from KPFM images of AZCS with or without light. **h** Surface potential from KPFM images of CZCS with or without light. **i** SPV spectra of AZCS and CZCS.

Charge carrier separation and transfer properties play a pivotal role in the photocatalytic kinetic process. Electrochemical impedance spectroscopy (EIS) spectra and photocurrent responses were obtained to probe the charge transfer resistance and charge carrier density. As shown in Fig. 4f, the EIS plots were fitted using an equivalent circuit (inset in Fig. 4f). Only one semicircle exists in the EIS curves of AZCS and CZCS, indicating that the interfacial exchange of charge carrier between the photocatalyst and electrolyte is the main impediment for carrier transport. Moreover, a smaller diameter in the EIS curve of AZCS compared to that of CZCS reflects the lower charge transfer resistance ($R_{ct}$). Supplementary Fig. S21 shows the diameter changes that are conform to the performance regular pattern of photocatalytic hydrogen evolution. The $R_{ct}$ values of all samples are shown in Supplementary Table S4, AZCS shows the lowest $R_{ct}$ of 62.28 kΩ, which is only 8% that of CZCS, suggesting the much faster surface charge transfer kinetics. Photocurrent responses under AM 1.5 G illumination of AZCS and CZCS are revealed in Fig. 4g. The AZCS shows over 23 times higher photocurrent densities than that of CZCS, suggesting a significantly higher hole-electron separation efficiency. Photocurrents of all samples with different crystallinities are presented in Supplementary Fig. S22. It can be observed that the photocurrent changes exhibit the same trend with their H₂ evolution performances, indicating the high charge separation efficiency in the sample with low crystallinity.

In a photocatalytic process, some of the photogenerated electrons are used for reduction reaction, while others unfortunately recombine with holes. The radiative recombination process was studied by photoluminescence (PL) spectroscopy. As demonstrated in Fig. 4h, under an excitation wavelength of 320 nm, AZCS shows a weaker band-to-band emission peak at 644 nm than its CZCS counterpart. The PL quenching result could be explained by the inhibition of charge recombination from the disorder atomic arrangements[56], as evidenced by much stronger PL peaks of ZCSRT, ZCS200 and ZCS400 compared to ZCS0 shown in Supplementary Fig. S23. To further elucidate the reaction dynamics of the excited charges, the emission decay profiles of AZCS and CZCS were recorded with $\lambda_{exc}$ = 375 nm by nanosecond time-resolved fluorescence (TR-PL). As shown in Fig. 4i, the PL lifetime (amplitude average lifetime) of AZCS is 12.8 ns, which is around 4.5 times higher than its CZCS counterpart (2.8 ns). The quenching is ascribed to the high recombination rate of the photogenerated electron-hole pairs in CZCS. The PL lifetime values of ZCSRT, ZCS200 and ZCS400 are 8.2, 5.3 and 3.8 ns, respectively (Supplementary Fig. S24), further indicating that the lower the crystallinity of ZnCdS, the higher the charge carrier transport efficiency.

**Dipole field characterizations**

To further verify the mechanism of the high photogenerated carrier separation efficiency in the amorphous structure, Kelvin probe force microscopy (KPFM) was applied to detect the surface potential profiles of AZCS and CZCS in dark and light under a Xe lamp illumination. As demonstrated in Fig. 5a, b, AZCS and CZCS were made into slurries and drip-coated on glass substrates for KPFM testing. The KPFM images of AZCS exhibit completely different surface potentials under dark and light conditions (Fig. 5c, d), and the surface potential does not vary uniformly according to the locations, which is caused by the dipole field generated by the amorphous structure at the surface. By contrast, the KPFM images of CZCS in Fig. 5e, f also present different surface potentials under dark and light, while the potential is varied uniformly

with the change of position. Furthermore, AZCS with light illumination exhibits an increase average surface potential of ~74.75 mV (Fig. 5g) compared to that without light, while its CZCS counterpart only shows an increase of ~49.42 mV (Fig. 5h), which further demonstrates that dipole field exists in AZCS.

The surface photovoltage (SPV) spectra were further used to reveal the surface charges of AZCS and CZCS. As shown in Fig. 5i, the positive signal implies that holes transmit to the illumination side of the samples, which is a representative property for n-type semiconductors under light irradiation. Moreover, the SPV of CZCS was scarcely detected, indicating the low charge separation and transportation, while AZCS obtains a significantly higher SPV signal and wider photo-response range. It is reasonable to state that the photocatalytic reaction is more likely to occur due to the migration of more photogenerated carriers to the AZCS surface with disordered arrangement of atomic structures[57].

We further found that the construction of an amorphous structure is a generic strategy to boost the photocatalytic $H_2$ evolution performance of metal sulfide photocatalysts. Amorphous and crystalline ZnS and CdS were synthesized by the same method, which are denoted as AZS, CZS, ACS and CCS, respectively. As shown in Supplementary Fig. S25, the TG curves of ZnS and CdS suggest that they are not decomposed at the temperature of 600 °C. The XRD and $H_2$ evolution of ACS and CCS are demonstrated in Supplementary Fig. S26. The amount of $H_2$ evolution of ACS is over 8 times higher than its CCS counterpart within 2.5 h (Supplementary Fig. S26a). The XRD peaks of CCS are sharp and conform to the peak of CdS (JCPDS-ICDD: 97-015-4186) while the XRD pattern of ACS only exhibits two envelope peaks (Supplementary Fig. S26b). Similarly, the $H_2$ evolution and XRD pattern of AZS and CZS are shown in Supplementary Fig. S27. Not surprisingly, AZS has better $H_2$ evolution properties than its CZS counterpart. The similar results of ZnS, CdS and ZnCdS illustrate the universality of the construction of amorphous structures to break the symmetric atomic arrangement in the crystal structure, inducing strong dipole field to promote bulk charge separation and transport, thereby significantly enhancing the photocatalytic performance.

## Discussion

A high-performance AZCS is synthesized by a facile low-temperature wet-chemical method. Owing to the asymmetry of the atom arrangement, strong dipole fields in the x-, y-, and z-directions are induced, providing extra driving forces for the separation and transport of the photogenerated electron-hole pairs. In addition, the bandgap of AZCS is significantly reduced from 2.67 to 2.34 eV because of the localization of band tail states in the amorphous structure. With the significantly enhanced bulk charge separation efficiency and the light utilization range, AZCS exhibits a $H_2$ evolution rate of 70.13 mmol $g^{-1}$ $h^{-1}$, which is over 5 times higher than its crystalline counterpart. An AQY of 38.54% is achieved at a monochromatic light with a wavelength of 420 nm. By engineering the surface $Co-MoS_x$ cocatalyst, the $AZCS/Co-MoS_x$ demonstrates a photocatalytic stability up to 160 h. A flexible photocatalytic film can be obtained by a facile blade-coating technique, demonstrating obvious $H_2$ evolution under natural sunlight illumination. The findings demonstrated in this work provide insights for the enhanced photocatalytic performance of amorphous photocatalysts, which may inspire the design of high-performance photocatalysts for solar fuel production.

## Methods

### Materials preparation

All of the chemical reagents were purchased from Beijing Innochem Science & Technology Co., Ltd. The amorphous ZnCdS (AZCS) was prepared by a low-temperature wet-chemical method. Briefly, 20 mL of aqueous $Na_2S·9H_2O$ solution (0.1 M, Aladdin, AR) was added into 20 mL of a mixed solution containing 0.05 M of $Zn(CH_3COO)_2·2H_2O$

(Innochem, 99%) and 0.05 M of $Cd(CH_3COO)_2·2H_2O$ (Aladdin, AR) with a rate of 1 mL/min and stirred at 0 °C for 1 h. The resulted yellow slurry was then washed thoroughly with Milli-Q water, and vacuum freeze-dried. ZCSRT was obtained by the same procedure of AZCS with the only change of the temperature to room temperature during stirring. ZCS200, ZCS400 and ZCS600 (ZCS600 is the most crystalline sample that is also denoted as CZCS) with higher crystallinity were synthesized by annealing the ZCSRT at 200, 400, and 600 °C in a tube furnace for 5 min under a nitrogen atmosphere, respectively. In order to examine the influence of different proportions of Zn and Cd, the samples of $Zn_xCd_{1-x}S$ (x = 0, 1/5, 2/5, 1/2, 3/5, 4/5, 1) were synthesized by the same conditions of AZCS.

Flexible $AZCS/Co-MoS_x$ films were prepared by a blade coating method. Briefly, 5 g of AZCS, 14 mg of $Co(NO_3)_2·6H_2O$ (Innochem, 99%), 50 mg of $(NH_4)_2MoS_4$ (Innochem, 99.95%) and 0.5 g of Poly (vinylidene fluoride) (Innochem, average Mw ~ 275,000 pellets) powder were added into a mortar. Then, 6 mL of 1-Methyl-2-pyrrolidinone (Innochem, 99.5%, extra dry) as the solvent was mixed in the above mixture and ground thoroughly for 30 minutes to obtain a slurry. An appropriate amount of the slurry was placed on aluminum foils with dimensions of 5 × 5 cm, 10 × 10 cm, and 20 × 20 cm, respectively, and the coating was scraped in one direction with a glass rod at a uniform speed to obtain uniform films. The obtained films were then dried in a vacuum oven at 60 °C for 1 h. The mass loadings of the films with dimensions of 5 × 5 cm, 10 × 10 cm, and 20 × 20 cm were 128.9, 826.1, and 3578.8 mg of $AZCS/Co-MoS_x$, respectively.

### Characterization

SEM analysis was carried out on a Zeiss Gemini 300 field emission scanning electron microscope. TEM, HRTEM, and SAED patterns were obtained on a FEI TalosF200x transmission electron microscope. XRD patterns were recorded on an X-ray (D8 Advance, Bruker) diffractometer with Cu $K_\alpha$ ($\lambda$ = 0.15406 nm) radiation. The crystallinities of the samples were calculated by the formula of $W_c = I_c / (I_c + I_a)$[58], where $I_c$ and $I_a$ represent the integrated area of the diffraction peak in the crystalline and amorphous states, respectively. Raman spectra were obtained on an Alpha 300 R Micro confocal Raman spectrometer using an TEM00 laser (532 nm). FTIR spectra were acquired on a Thermo Scientific Nicolet iS5 spectrometer. XPS and VB XPS spectra were obtained on an XPS scanning microprobe spectrometer with an Al $K_\alpha$ ($h\nu$ = 1253.6 eV) radiation source. C 1$s$ (284.8 eV) was used as a reference to calibrate the binding energies. The Zn, Cd and S ion concentrations were analyzed by an Agilent 5110 inductively coupled plasma optical emission spectrometry (ICP-OES). Nano particle distribution was detected by a nanoparticle size analyzer (Malvern Zetasizer Nano ZS90). UV-vis absorbance spectra of the solid powder samples were recorded on a spectrophotometer (UV-2600i, Shimadzu). PL measurements were carried out on a fluorescence spectrophotometer (FLS-1000, Edinburgh Instruments) with an excitation light at 375 nm. KPFM images were obtained by KPFM mode for Atomic Force Microscopy (Bruker Dimension Icon). SPV spectra were obtained on a CEL-SPS1000 surface photovoltage spectrometer.

### Photocatalytic tests

Photocatalytic $H_2$ evolution was conducted at 6 °C in a photocatalytic activity evaluation system (Beijing China Education Au-Light Co., Ltd., CEL-PAEM-D8) according to our previous study[54]. A gas chromatography (Beijing China Education Au-Light Co., Ltd., GC-7920) was equipped with a thermal conductive detector (TCD) and a TDX-01 molecular sieve column. A 300 W Xenon lamp (350 nm <λ < 780 nm) was employed as the light source of simulated solar irradiation. Argon was used as the carrier gas. 20 mg of photocatalysts were dispersed in 60 mL of aqueous solution containing 10 mL of lactic acid (Aladdin, 90%). The $Co-MoS_x$ cocatalyst was decorated on the photocatalyst surfaces by photo-deposition with $Co(NO_3)_2·6H_2O$ (Innochem, 99%)

and $(NH_4)_2MoS_4$ (Innochem, 99.95%) aqueous solution equivalent to 1 wt.% for 10 min before photocatalytic reactions. The produced gas was analyzed every 30 min automatically controlled by the software.

The apparent quantum yield (AQY) test method for hydrogen evolution at different wavelengths is similar to the hydrogen evolution test but only with monochromatic illumination which was calculated as follows: AQY (%) = $N_e/N_p \times 100\%$ = $(2 M \times N_A \times H \times c)/(S \times P \times t \times \lambda) \times 100\%$[59], where $N_e$ refers to the reacted electrons and $N_p$ represents the number of incident photons. $M$ is the amount of evolved $H_2$ molecules (mol), $N_A$ is the Avogadro constant, $h$ is the plank constant, and $c$ represents the speed of light. $S$ is the irradiation area (19.6 $cm^2$), $P$ is the power density of the incident light, $t$ is the irradiation time, and $\lambda$ is the wavelength of the monochromatic light.

## Photoelectrochemical measurements

5 mg of the samples were distributed in a solution containing 0.375 mL of $H_2O$, 0.125 mL of ethanol (Greagent, 99.7%) and 50 µL of Nafion (Alfa, 5% in water and 1-propanol) by ultrasonication for 1 h. 50 µL of the slurry was deposited on a clean fluorine doped $SnO_2$ (FTO) glass substrate (2 × 1.5 cm) by spin-coating. After drying on a hot place at 60 °C, the obtained samples were used as the working electrode and the exposed area of the active material on the working electrode was controlled as 1 × 1.5 cm. Photoelectrochemical (PEC) performance were measured in a typical three-electrode cell under the illumination of a Xe lamp equipped with an AM 1.5 G filter. A platinum wire served as the counter electrode, and an Ag/AgCl (saturated KCl) electrode was applied as the reference electrode. During the test, the active area was controlled as 1 $cm^2$. Photocurrent responses were obtained at the circuit potential of the electrode under 100 mW $cm^{-2}$ illumination for 400 s with an interval of 50 s every 50 s in a 0.5 M $Na_2SO_4$ electrolyte. EIS spectra were conducted with an AC voltage amplitude for 0.1 V at the open circuit potentials of the electrode in dark (frequency range: 0.01 ~ 200 kHz) using 0.5 M $Na_2SO_4$ as the electrolyte. MS plots were obtained at a voltage range of −0.32 to 0.67 V versus NHE (frequency: 1200 Hz) in dark using 0.5 M $KH_2PO_4$ electrolyte (the pH was adjusted to 7 by KOH).

## DFT calculations

All calculations were performed using the density functional theory (DFT), as implemented in the Vienna ab initio simulation package[60,61]. The projector augmented-wave (PAW) method and Perdew-Burke-Ernzerhof generalized gradient approximation (GGA-PBE) were used for the exchange correlation functionals[62,63]. The time step was set to 3 fs and only the $\Gamma$ point was sampled from the Brillouin zone. The canonical (NVT) ensemble with the Nose−Hoover thermostat was applied to control the temperature and the pressure in AIMD simulations. Crystalline ZnCdS was a hexagonal crystal system, and the unit cell model was obtained based on the XRD results. The initial supercell contained 192 atoms of ZnCdS (4 × 4 × 3) with a cell parameter >16 Å. The amorphous models were obtained by using the melt-quenched process: Firstly, the primitive models were fully melted at 3000 K to remove the memory effect from the initial sites. Secondly, the models were cooled down to 1400 K and relaxed at this temperature for a stable liquid. Then, we rapidly reduced the temperature to 300 K with a cooling rate of 33.3 K/ps, and finally, the models were maintained at 300 K for 30 ps to collect the trajectories of the atoms. For the calculations of electronic structure, the energy cutoff of the PAW basis was set to 450 eV with a force convergence of 0.02 eV and a 2 × 2 × 1 $k$-point grid was selected for the Brillouin zone sampling. The $U_{eff}$ values of Zn and Cd were calculated based on the bandgap of ZnCdS. When the $U_{eff}$ values of Zn and Cd were higher than 7.0 and 4.0 eV, the bandgap ZnCdS trends to be stable. Therefore, the Coulomb interaction $U_{eff}$ values were set to 7.0 and 4.0 eV to describe the 3$d$ electrons

of Zn and 4$d$ electrons of Cd, respectively, which are also consistent to the literature[64,65].

## Data availability

The data generated within the paper and its Supplementary Information file are available from the corresponding authors upon request. Source data of Figs. 2o–r, 3a–e, 4a–d, f–i, and 5g–i, and Supplementary Figs. S2–S27 are provided in a Source Data file. Source data are provided with this paper.

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

## Acknowledgements

This work was financially supported by the National Natural Science Foundation of China (No. 52372292), and the Fundamental Research Funds for the Central Universities. The authors would also like to acknowledge financial support and material characterization from the Analytical & Testing Center of Northwestern Polytechnical University (No. 2022T016), and the SPV spectra measurement from Beijing China Education Au-Light Co., Ltd. L.W. is thankful for the financial support of the Australian Research Council through its Discovery Project (DP200101900) and Laureate Fellowship (FL190100139).

## Author contributions

S.W. guided and designed the project. X.W. carried out the experiments and wrote the manuscript. B.L. and S.M. fabricated electrodes for PEC measurements and performed the characterization. Y.Z. conducted SEM characterization. X.W., S.W., L.W., G.Z., and W.H. analyzed the data and prepared the manuscript. All the authors commented and approved the paper.

## Competing interests

The authors declare no competing interests.
