## [Peer Review File · Nature Communications]

Induced dipole moments in amorphous ZnCdS catalysts facilitate photocatalytic H₂ evolutionREVIEWER COMMENTS

Reviewer #1 (Remarks to the Author):

This manuscript firstly conducted theoretical simulations to predict the induction of dipole moments in amorphous ZnCdS (AZCS) photocatalyst, which can provide extra driving forces to promote charge separation and transfer. Then, a facile wet-chemical process was developed to prepare the AZCS sample. With the loading of a low-cost Co-MoS_x cocatalyst, a record-breaking hydrogen production rate is achieved. Importantly, an excellent stability of 160 h is achieved, which is very significant for sulfide photocatalysts. It is also exciting to see that a flexible photocatalytic film with dimensions of 20 cm × 20 cm demonstrates obvious photocatalytic hydrogen production under natural sunlight. The work is interesting and should have a high impact in the field of solar hydrogen production and related fields. Thus, I recommend the acceptance of this work for publication after revision. The main concerns are as follows:

1. In Page 7, according to the TEM images, "AZCS is composed of connected nanoparticles with a size of approximately 42.4 nm" "numerous nanoparticles with a size of around 49.1 nm is observed in CZCS". However, in Page 8, the authors claimed that "As demonstrated in Supplementary Fig. S1, the particle size distribution of AZCS is 100-250 nm, while that of CZCS is 100-350 nm." Why the particle sizes of AZCS and CZCS shown in the TEM images are different from those in Supplementary Fig. S1? The authors should provide more explanations.
2. In Fig. 5i, the unit of photovoltage (e.g. mV or V) should be provided, so that readers can better understand the photovoltage difference between AZCS and CZCS.
3. To better understand the underlying mechanism of the stability shown in Figure 3c, it is suggested to do more material characterizations of the photocatalyst after the stability test.
4. Details of preparing the flexible Co-MoS_x/AZCS films should be provided.
5. For PEC analysis, more details of the dimensions of the FTO substrates and the exposed area of the prepared photoelectrodes should be provided.

Reviewer #2 (Remarks to the Author):

This work presents a computational and experimental study which claimed that the amorphous configuration in ZnCdS (AZCS) photocatalyst, when loaded with Co-MoS_x cocatalyst, can show improved H₂ evolution efficiency. The reason was attributed to the lattice asymmetry induced dipole moments, which provide extra driving forces to promote charge separation and transfer. It reads well written and organized, and many characterizations have been carried out. However, I am concerned as to whether the fivefold improvement in activity can be attributed entirely to the dipole effect, as I have seen no more direct evidence. I cannot recommend its publication in the current content.

(i) In the presence of disordered configurations, the nature of the cocatalyst is also altered by tuning the interfacial binding, which could significantly change the intrinsic catalytic ability. So how can the authors decouple this effect?

- (ii) It is important to discuss that how the observed trend may vary if the Co-MoS_x cocatalyst is substituted with another.
- (iii) In my opinion, the amorphous configuration is common in many synthesized catalysts and the dipole effect is not a new thing. As for the presence of amorphous configurations, it is not always considered to be conducive to the charge separation, as also mentioned by the authors in the introduction. Then, how is ZnCdS unique to guarantee the charge separation while others are not.
- (iv) The current DFT calculation results shown in Figure 1 are not convincing. The amorphous structure in Figure 1b looks completely unreasonable with one- and three-coordinated Cd exposed. Thus, the current theoretical conclusions (including the dipole calculations) lack of credibility and reference value.
- (v) Fig. 1f and 1h are only a conceptual demonstration and there is no actual data to support them. Also, I cannot understand the Fig. 1h on the energy band bending, which lacks definition and looks strange and even misleading.
- (vi) From the numerical data in Fig. 1g, the dipole direction is completely opposite to the surface exposure (Z) direction, which in principle would change the relative distribution of photoexcited holes and electrons and greatly affect the H₂ evolution efficiency and even change the trend. This is not consistent with the only fivefold improvement in activity.

Reviewer #3 (Remarks to the Author):

This paper introduces a novel amorphous ZnCdS (AZCS) photocatalyst that exhibits remarkable photocatalytic hydrogen evolution performance. The authors attribute this to the asymmetric atom arrangement in AZCS, which induces a strong dipole field that enhances charge separation and transfer. Density functional theory (DFT) calculations are used to elucidate the distinct electronic structure features of crystalline ZnCdS (CZCS) and AZCS materials. However, this paper has some critical problems in both calculation and experimentation that need to be addressed. Only after these issues are properly resolved, can this work be considered for publication on Nature Communications.

1. The author should clarify the nature of the AZCS synthesized in their experiment. Is it truly amorphous or rather polycrystalline? In Fig. 2c, it shows some ordered regions in AZCS with a size of several nanometers. Moreover, the electron diffraction patterns and the XRD results in Fig. 2o suggest that the sample is more likely to be polycrystalline with small particle size rather than amorphous. The broadening of the two main peaks in the XRD spectrum could be explained by the Scherrer's formula for polycrystalline particles. On the other hand, amorphous samples usually have only one large broad peak in the XRD spectrum. Therefore, the author may need to reevaluate the structural model of the material.
2. The authors claim that the amorphous structure induces strong dipole fields that promote charge separation. However, this argument is not convincing for two reasons. First, the structural model of the amorphous ZnCdS is not well supported by experimental evidence. Second, the dipole changes are calculated from small super cell model contains only two periodic layers, which is too simplistic to

capture the complex disorder and local variations in the amorphous structure.

3. In the manuscript, the authors did not explain how the structural models of CZCS and AZCS were constructed for the DFT calculation. In particular, the details of the AZCS model is unclear. According to Fig. 1, the AZCS model seems to be derived from the CZCS crystal model by introducing local distortions, but still retains the alternating layered structure of ZnS and CdS. This contradicts the “random arrangement” of [ZnS₄] and [CdS₄] units that the author claimed in the main text (line 88, page 5).

4. The DFT calculated energy of AZCS and CZCS is not reported. The authors need to compare the stability of the theoretical models AZCS and CZCS and verify them with their experimental results.

5. The separation of photo generated electrons and holes in AZCS may be validated by the spatial distribution of the HOMO and LUMO orbitals.

6. In the Methods section, it lacks some key information in the calculation details, such as the size of the supercell and the construction method of the AZCS structural model. These should be clearly stated and justified.

7. The U values used in the calculations is important and could lead to inaccurate results. The authors should use Hubbard U and exchange parameter J or U_{eff} to represent the U values, and provide a detailed procedure of how they were tested and chosen.

Response to Reviewers' Comments

Reviewer #1 (Remarks to the Author):

This manuscript firstly conducted theoretical simulations to predict the induction of dipole moments in amorphous ZnCdS (AZCS) photocatalyst, which can provide extra driving forces to promote charge separation and transfer. Then, a facile wet-chemical process was developed to prepare the AZCS sample. With the loading of a low-cost Co-MoS_x cocatalyst, a record-breaking hydrogen production rate is achieved. Importantly, an excellent stability of 160 h is achieved, which is very significant for sulfide photocatalysts. It is also exciting to see that a flexible photocatalytic film with dimensions of 20 cm × 20 cm demonstrates obvious photocatalytic hydrogen production under natural sunlight. The work is interesting and should have a high impact in the field of solar hydrogen production and related fields. Thus, I recommend the acceptance of this work for publication after revision. The main concerns are as follows.

Reply: We are grateful to you for the positive feedbacks. We have revised the manuscript according to your comments.

1. In Page 7, according to the TEM images, “AZCS is composed of connected nanoparticles with a size of approximately 42.4 nm” “numerous nanoparticles with a size of around 49.1 nm is observed in CZCS”. However, in Page 8, the authors claimed that “As demonstrated in Supplementary Fig. S1, the particle size distribution of AZCS is 100-250 nm, while that of CZCS is 100-350 nm.” Why the particle sizes of AZCS and CZCS shown in the TEM images are different from those in Supplementary Fig. S1? The authors should provide more explanations.

Reply: Thank you for your valuable comments. We are sorry for making the confusion. Actually, the difference is from the different instruments for measuring the particle size. The nanoparticle size obtained from TEM is only from several particles shown in the TEM image. In addition, we only manually measured the observable particles, and the aggregation of the particles is not taken into account.

In comparison, the nanoparticle size distribution obtained in Supplementary Fig. S4 (Supplementary Fig. S1 in the previous version) is from the test results of a nanoparticle

size analyzer, which is measured as the equivalent volume particle size. The sample is dispersed in a liquid for measurement. When the particles are pushed by the thermal motion of the medium molecules in the liquid to undergo Brownian motion, the medium molecules are carried and attached to the surface of the particles. The medium molecules move together, so the particle size measured by a nanoparticle size analyzer is actually the diameter of the particle plus twice the thickness of medium molecules. In addition, aggregation of the nanoparticles may lead to the movement of several particles together as a whole. Therefore, the particle size measured by a nanoparticle size analyzer is much larger than its counterpart measured by TEM. Although the test results of these two methods are different, the comparison of two different samples measured by the same method is reliable.

To avoid any misunderstanding of the particle size values in our manuscript, two more sentences have been added in the main text for further explanation, as highlighted in Page 9-10 in the revised manuscript.

Changes to the revised manuscript are shown below.

Main Manuscript (Results):

Page 9-10: It should be mentioned that the particle size measured by a nanoparticle size analyzer is the statistical results from 20 mg of the sample, while the particle size measured by TEM is only the observable particles shown in the TEM image. Therefore, the particle size values measured by these two different methods may be different. However, the particle sizes of different samples measured by the same method can be reasonably compared.

2. In Fig. 5i, the unit of photovoltage (e.g. mV or V) should be provided, so that readers can better understand the photovoltage difference between AZCS and CZCS.

Reply: Thank you for your suggestions. We apologize for the oversight in not including the unit of the photovoltage in the manuscript. The relevant figure is now provided in our revised manuscript, as highlighted in Page 23 in the main text. We have attached the revised figure as Fig. R1 below for your information.

Fig. R1. SPV spectra of AZCS and CZCS (Fig. 5i).

3. To better understand the underlying mechanism of the stability shown in Figure 3c, it is suggested to do more material characterizations of the photocatalyst after the stability test.

Reply: Thank you very much for your constructive suggestions to further improve the quality of our manuscript. We totally agree that it is important to do more material characterizations of the photocatalyst after stability test, which is very helpful to understand the mechanism. We supplemented the SEM and XRD characterizations of four samples: AZCS without Co-MoS_x loading after 8 h of photocatalysis (denoted as AZCS-8h), AZCS photo-deposited with Co-MoS_x before photocatalysis measurement (denoted as AZCS/CMS), AZCS with Co-MoS_x added in each cycle during 5 cycles of photocatalytic reaction (denoted as AZCS/5-CMS), and AZCS with Co-MoS_x added only in the first cycle after 5 cycles of photocatalytic reaction (denoted as AZCS/CMS-5). SEM images shown in Fig. R2 (Supplementary Fig. S18) indicate that their morphologies and particle sizes are almost the same. In addition, XRD patterns of the sample shown in Fig. R3 (Supplementary Fig. S19) are also very similar. Therefore, after the stability test of photocatalysis, the morphology and structure of the AZCS sample remain unchanged, indicating the excellent stability of the sample. To clarify this, we have illustrated the detail in the revised manuscript.

Fig. R2. SEM images of AZCS-8h, AZCS/CMS, AZCS/5-CMS and AZCS/CMS-5 (Supplementary Fig. S18).

Fig. R3. XRD spectra of AZCS-8h, AZCS/CMS, AZCS/5-CMS and AZCS/CMS-5 (Supplementary Fig. S19).

Changes to the revised manuscript are shown below.

Main manuscript (Results):

Page 16-17: SEM, XRD and ICP were used to characterize the mechanism of Co-MoS_x in promoting the photocatalytic stability of AZCS, and the results demonstrate that the morphology, structure, and elements in AZCS are relatively stable after photocatalytic H₂ evolution test (Supplementary Figs. S18, 19 and Supplementary Table S3, Supplementary Discussion).

Supplementary Information:

Page 15, Supplementary Discussion

To further understand the underlying mechanism for the excellent photostability, SEM and XRD were used to characterize the changes in morphology and structure of AZCS without Co-MoS_x loading after 8 h of photocatalysis (denoted as AZCS-8h), AZCS photo-deposited with Co-MoS_x before photocatalysis measurement (denoted as AZCS/CMS), AZCS with Co-MoS_x added in each cycle during 5 cycles of photocatalytic reaction (denoted as AZCS/5-CMS), and AZCS with Co-MoS_x added only in the first cycle after 5 cycles of photocatalytic reaction (denoted as AZCS/CMS-5). As shown in the Figs. S18, 19, no obvious change can be observed in both SEM images and XRD patterns, indicating the excellent stability of our newly developed AZCS photocatalyst.

Page 23, Supplementary Discussion

To further understand the change of Co-MoS_x cocatalyst on AZCS with and without additional Co-MoS_x cocatalyst in each cycle during photocatalytic hydrogen production, ICP was used to investigate the amount of cocatalyst in AZCS/Co-MoS_x and the supernatant liquid after each hydrogen evolution cycle. As listed in Supplementary Table S3, with additional Co-MoS_x cocatalyst in each cycle, the Mo content in AZCS/Co-MoS_x increases after 1, 5, and 10 cycles, and the supernatant liquid after 1 and 10 cycles also contains a small amount of Mo. However, without additional Co-MoS_x cocatalyst in each cycle, almost no Mo signals can be detected in the supernatant liquid after photocatalytic hydrogen evolution for 5 cycles. In addition, the Cd contents in the photocatalyst and the supernatant liquid are almost unchanged from the 1st to the 10th cycle, indicating that AZCS itself is stable during the photocatalytic process. The above results indicate that not all Co-MoS_x are attached to AZCS during the photo-

deposition process, and the loading amount of Co-MoS_x on AZCS is relatively stable during photocatalytic hydrogen evolution. However, the Co-MoS_x cocatalyst gradually lose activity for H₂ evolution (Supplementary Fig. S16). During the photocatalytic process, the reducible support easily forms overlayers on the surface of the cocatalyst, thereby affecting the redox reaction on the surface of the photocatalyst^{31,32}. Therefore, adding an appropriate amount of Co-MoS_x cocatalyst during the photocatalytic process is necessary to maintain the activity for hydrogen evolution, thereby achieving excellent long-term stability.

4. Details of preparing the flexible Co-MoS_x/AZCS films should be provided.

Reply: Thank you for your suggestion. We apologize for the oversight of not providing a complete description of the preparation of the flexible Co-MoS_x/AZCS films. We have updated the methodology to cover more details in the revised manuscript.

Changes to the revised manuscript are shown below.

Main Manuscript (Methods – Materials preparation)

Pages 26: Flexible AZCS/Co-MoS_x films were prepared by a blade coating method. Briefly, 5g of AZCS, 14 mg of Co(NO₃)₂·6H₂O, 50 mg of (NH₄)₂MoS₄ and 0.5 g of poly(vinylidene fluoride) powder were added into a mortar. Then, 6 mL of 1-Methyl-2-pyrrolidinone (99.5%, extra dry, Innochem) as the solvent was mixed in the above mixture and grinded thoroughly for 30 minutes to obtain a slurry. An appropriate amount of the slurry was placed on aluminum foils with dimensions of 5 cm × 5 cm, 10 cm × 10 cm, and 20 cm × 20 cm, respectively, and the coating was scraped in one direction with a glass rod at a uniform speed to obtain uniform films. The obtained films were then dried in a vacuum oven at 60 °C for 1 h. The mass loadings of the films with dimensions of 5 cm × 5 cm, 10 cm × 10 cm, and 20 cm × 20 cm were 0.1289, 0.8261, and 3.5788 g of AZCS/Co-MoS_x, respectively.

5. For PEC analysis, more details of the dimensions of the FTO substrates and the exposed area of the prepared photoelectrodes should be provided.

Reply: We appreciate your excellent suggestions. We apologize for the oversight in not providing a complete description of the PEC analysis. We have updated the methods to

include more details, including FTO dimensions and the exposed area of the photoelectrode in the revised manuscript.

Changes to the revised manuscript are shown below.

Main Manuscript (Methods – Photoelectrochemical measurements)

Page 28: 50 μL of the slurry was deposited on a clean fluorine doped SnO_2 (FTO) glass substrate ($2\text{ cm} \times 1.5\text{ cm}$) by spin-coating. After drying on a hot place at $60\text{ }^\circ\text{C}$, the obtained samples were used as the working electrode and the exposed area of the active material on the working electrode was controlled as $1\text{ cm} \times 1.5\text{ cm}$.

Reviewer #2 (Remarks to the Author):

This work presents a computational and experimental study which claimed that the amorphous configuration in ZnCdS (AZCS) photocatalyst, when loaded with Co-MoS_x cocatalyst, can show improved H₂ evolution efficiency. The reason was attributed to the lattice asymmetry induced dipole moments, which provide extra driving forces to promote charge separation and transfer. It reads well written and organized, and many characterizations have been carried out. However, I am concerned as to whether the fivefold improvement in activity can be attributed entirely to the dipole effect, as I have seen no more direct evidence. I cannot recommend its publication in the current content.

Reply: We are grateful to you for providing valuable insights that have significantly contributed to the refinement and enhancement of our manuscript. We have carefully considered your comments and acknowledge the concerns raised regarding the attribution of the observed activity improvement solely to the dipole effect.

It is true that the photocatalytic activity is affected by many factors. Based on the systematic studies in our manuscript, we would like to emphasize the key difference between CZCS and AZCS. We first applied DFT calculations to investigate the difference of dipole moments between CZCS and AZCS, and the results confirmed that the dipole moments along the *x*-, *y*- and *z*-directions in AZCS are significantly higher than those in CZCS (Fig. 1e). In addition, we also applied Kelvin probe force microscopy (KPFM) and surface photovoltage (SPV) spectra to characterize the surface potential induced by light illumination (Fig. 5), which is the experimental evidence to confirm the presence of dipole fields in AZCS. It should be mentioned that since the

photocatalytic activity is affected by many factors (e.g., light absorption, charge separation and transfer, surface photocatalytic reactions, etc.), we would like to take a systematic analysis on the key factors below.

According to our studies in Fig. 4a, b, the bandgap of AZCS (2.34 eV) is smaller than its CZCS counterpart (2.67 eV), which means that AZCS has a stronger light absorption capability. However, the difference of 0.33 eV in bandgap between the two sample cannot cause a difference of about 5 times in the photocatalytic H₂ evolution rate. To drive a photocatalytic reaction, both thermodynamics and kinetics should be taken into account. For example, the bandgap of the 2DPA sample reported in *Nat. Commun.*, 2018, 9, 4036 is larger than that of 2DA. However, the photocatalytic H₂ evolution performance of 2DPA is higher than that of 2DA.

In order to exclude the effect of surface cocatalysts, we also measured the photocatalytic H₂ evolution performances of CZCS and AZCS without any cocatalyst. As shown in Fig. R5 (please refer to our response in your valuable Comment 1 for more information), the photocatalytic H₂ evolution rate of AZCS is also about 5 times higher than its CZCS counterpart. Therefore, the photocatalytic activity improvement in AZCS is originated from the structure itself, rather than the surface loaded cocatalysts.

After excluding the factors of light absorption and surface redox reactions, we focused on the separation and transmission rate of photogenerated carriers. From electrochemical studies (EIS, I-t) and PL characterization, we concluded that AZCS has stronger charge transport capability. Since CZCS and AZCS have different carrier separation efficiencies under the same photocatalytic conditions, we believe that the carrier transport driving force is caused by their different internal structures. The existence of this driving force is also directly characterized by KPFM and surface photovoltage. Based on the DFT calculation and experimental results, we believe it is reasonably to state that the dipole effect has the highest contribution to the improvement of photocatalytic performance. We once again express our appreciation for your excellent comments to further improve the quality of our manuscript.

We have taken all your excellent comments into account in our revised manuscript. Below, please read our careful response to your comments point-by-point.

1. In the presence of disordered configurations, the nature of the cocatalyst is also altered by tuning the interfacial binding, which could significantly change the intrinsic catalytic ability. So how can the authors decouple this effect?

Reply: Many thanks for your insightful comments. This is a very good point that worth further investigations. We have applied Raman spectroscopy to characterize the influence of the Co-MoS_x cocatalyst on the vibration modes of AZCS and CZCS functional groups, respectively. Fig. R4 (Supplementary Fig. S9) shows that although Co-MoS_x does not exhibit the characteristic peaks on the Raman spectrum, it will affect the characteristic peaks of AZCS and CZCS, respectively. Almost no second longitudinal-optical phonons (2LO) appear in AZCS/Co-MoS_x while AZCS has 2LO characteristics, indicating that Co-MoS_x can suppress the 2LO characteristics of AZCS. However, it is worth noting that the intensity of the 2LO characteristic peak in CZCS/Co-MoS_x also becomes weaker. Therefore, loading Co-MoS_x cocatalyst will affect the vibration mode of the characteristic functional groups of both AZCS and CZCS.

In order to understand the influence of the cocatalyst on the photocatalytic H₂ evolution performance of AZCS, we have measured the photocatalytic H₂ evolution performances of AZCS and CZCS with different cocatalysts (e.g. MoS₂ and Pt) and without cocatalyst (as demonstrated in Figs. R5, 6). Interestingly, AZCS and CZCS without cocatalyst also showed photocatalytic H₂ evolution performance. Within the first half an hour, the ratio of the photocatalytic H₂ evolution rates between AZCS and CZCS is 5.11. Although the photocatalytic H₂ evolution activity of AZCS and CZCS with different cocatalysts are different, it is observed no matter what kind of cocatalyst is applied (or even without cocatalyst), the photocatalytic H₂ evolution rate of AZCS is over four times higher than its CZCS counterpart. This can also indirectly prove that AZCS and CZCS have intrinsic characteristics that make them different in photocatalytic performance, and the cocatalyst does not play a pivotal role in affecting the relative activity between AZCS and CZCS.

Fig. R4. Raman spectra of AZCS, AZCS/Co-MoS_x, CZCS and CZCS/Co-MoS_x (Supplementary Fig. S9).

Fig. R5. Photocatalytic H₂ evolution of AZCS and CZCS loaded with different Co-catalyst and without Co-catalyst. (Supplementary Fig. S10).

Fig. R6. The ratio of H₂ evolution produced by AZCS and CZCS loaded with different cocatalysts at different time points. (Supplementary Fig. S11).

Changes to the revised manuscript are shown below.

Main Manuscript (Results)

Page 13-14: To understand whether the interfacial binding of the photocatalyst and the cocatalyst will affect the photocatalytic activity, Raman spectra (Supplementary Fig. S9) of Co-MoS_x loaded on AZCS and CZCS, and the photocatalytic H₂ evolution performances of AZCS and CZCS loaded with different cocatalysts (Supplementary Figs. S10-12) were performed. The results prove that the cocatalyst does not affect the relative activity of AZCS and CZCS, while playing a pivotal role in accelerating surface photocatalytic reactions to alleviate the side reactions between the photocatalyst and the photogenerated charge carriers (Supplementary Discussion).

Supplementary Information

Page 8-9, Supplementary Discussion

Since the interfacial binding between the photocatalyst and cocatalyst may also affect the photocatalytic activity, Raman spectroscopy was used to characterize the effect of Co-MoS_x on the vibration modes of AZCS and CZCS functional groups, respectively. As shown in Supplementary Fig. S9, although Co-MoS_x does not show its characteristic peaks on the Raman spectrum, it will affect the characteristic peaks of AZCS and CZCS, respectively. Since the intrinsic structures of AZCS and CZCS are different, the influence of Co-MoS_x on their Raman peaks is also different. To exclude the effect of cocatalyst on the enhanced photocatalytic activity of AZCS compared to CZCS, the influence trend of different cocatalysts on the photocatalytic H₂ evolution performance of AZCS and CZCS was also investigated. As shown in Supplementary Figs. S10, 11, when Co-MoS_x, MoS₂ and Pt were used as the cocatalysts, the ratios of the average hydrogen evolution rates of AZCS and CZCS are 5.0, 4.7 and 4.03, respectively.

It is worth noting that without a cocatalyst, AZCS and CZCS also have photocatalytic H₂ evolution properties. In the first half an hour, the ratio of the hydrogen evolution rates of AZCS and CZCS is 5.11, indicating that AZCS and CZCS have an inherent difference in photocatalytic H₂ evolution performance. Owing to the sluggish photocatalytic reaction without a cocatalyst, the difference between the photocatalytic performance of AZCS and CZCS gradually decreases. It should be mentioned that the photocatalytic H₂ production rate of AZCS gradually decreases with the irradiation time, while its CZCS counterpart exhibits a relatively stable photocatalytic H₂ production rate during the same irradiation time, indicating that AZCS without cocatalyst is less

stable than CZCS. These experimental results are consistent with the DFT calculations shown in Supplementary Fig. S2.

As shown in Supplementary Fig. S12, the color of AZCS without a cocatalyst appears dark green after 8 h of photocatalysis, while its counterpart with a Co-MoS_x as cocatalyst is still yellow, indicating that cocatalyst is essential to accelerate surface photocatalytic reactions and thus alleviating the destroy of the photocatalyst itself by the photogenerated charge carriers.

2. It is important to discuss that how the observed trend may vary if the Co-MoS_x cocatalyst is substituted with another.

Reply: Many thanks for your excellent suggestions. We have updated the analysis of photocatalytic H₂ evolution experiments with MoS₂ and Pt used as cocatalysts and without any cocatalyst. As shown in Figs. R5, 6 (Supplementary Figs. S10, 11), the average H₂ evolution rate of AZCS/MoS₂ is 4.7 times higher than its CZCS/MoS₂ counterpart, while the average H₂ evolution rate of AZCS/Pt is 4.03 times higher than that of CZCS/Pt. It is worth noting that AZCS also exhibits photocatalytic H₂ evolution performance without any cocatalyst. In the first half an hour, the H₂ evolution rate of AZCS is 5.11 times higher than the CZCS counterpart. Although the photocatalytic H₂ evolution activity of AZCS and CZCS with different cocatalysts are different, it is observed no matter what kind of cocatalyst is applied (or even without cocatalyst), the photocatalytic H₂ evolution rate of AZCS is over four times higher than its CZCS counterpart. This can also indirectly prove that AZCS and CZCS have intrinsic characteristics that make them different in photocatalytic performance, and the cocatalyst does not play a pivotal role in affecting the relative activity between AZCS and CZCS.

Changes to the revised manuscript are shown below.

Main Manuscript (Results)

Page 13-14: To understand whether the interfacial binding of the photocatalyst and the cocatalyst will affect the photocatalytic activity, Raman spectra (Supplementary Fig. S9) of Co-MoS_x loaded on AZCS and CZCS, and the photocatalytic H₂ evolution performances of AZCS and CZCS loaded with different cocatalysts (Supplementary Figs. S10-12) were performed. The results prove that the cocatalyst does not affect the

relative activity of AZCS and CZCS, while playing a pivotal role in accelerating surface photocatalytic reactions to alleviate the side reactions between the photocatalyst and the photogenerated charge carriers (Supplementary Discussion).

Supplementary Information

Page 8-9, Supplementary Discussion

To exclude the effect of cocatalyst on the enhanced photocatalytic activity of AZCS compared to CZCS, the influence trend of different cocatalysts on the photocatalytic H₂ evolution performance of AZCS and CZCS was also investigated. As shown in Supplementary Figs. S10, 11, when Co-MoS_x, MoS₂ and Pt were used as the cocatalysts, the ratios of the average hydrogen evolution rates of AZCS and CZCS are 5.0, 4.7 and 4.03, respectively.

It is worth noting that without a cocatalyst, AZCS and CZCS also have photocatalytic H₂ evolution properties. In the first half an hour, the ratio of the hydrogen evolution rates of AZCS and CZCS is 5.11, indicating that AZCS and CZCS have an inherent difference in photocatalytic H₂ evolution performance. Owing to the sluggish photocatalytic reaction without a cocatalyst, the difference between the photocatalytic performance of AZCS and CZCS gradually decreases. It should be mentioned that the photocatalytic H₂ production rate of AZCS gradually decreases with the irradiation time, while its CZCS counterpart exhibits a relatively stable photocatalytic H₂ production rate during the same irradiation time, indicating that AZCS without cocatalyst is less stable than CZCS. These experimental results are consistent with the DFT calculations shown in Supplementary Fig. S2.

As shown in Supplementary Fig. S12, the color of AZCS without a cocatalyst appears dark green after 8 h of photocatalysis, while its counterpart with a Co-MoS_x as cocatalyst is still yellow, indicating that cocatalyst is essential to accelerate surface photocatalytic reactions and thus alleviating the destroy of the photocatalyst itself by the photogenerated charge carriers.

3. In my opinion, the amorphous configuration is common in many synthesized catalysts and the dipole effect is not a new thing. As for the presence of amorphous configurations, it is not always considered to be conducive to the charge separation, as also mentioned by the authors in the introduction. Then, how is ZnCdS unique to guarantee the charge separation while others are not.

Reply: Thank you for your valuable comments. It is true that amorphous configuration is reported in many publications in electrocatalysis (Adv. Energy Mater., 2022, 12, 2200827; Small, 2020, 16, 1905779; Adv. Mater., 2019, 31, 1900883). However, very few is reported in photocatalysis. Traditionally, amorphous materials are generally considered to have no photocatalytic activity due to their lack of long-range order and numerous defects (Adv. Mater., 2015, 27, 4572-4577). However, in recent years, some researchers found that amorphous semiconductors showed some interesting properties in photocatalysis. For example, the amorphous band tail state effect makes the bandgap of TiO₂ smaller and absorbs more visible light energy, which is called black TiO₂ (J. Am. Chem. Soc., 2008, 130,14755-14762; Science, 2011, 331, 746-750). In addition, Nat. Commun., 2018, 9, 4036 reported an amorphous NiO with photocatalytic effect, whose crystalline state usually acts as an electrocatalyst but does not have photocatalytic properties. Another example is the amorphous carbon nitride that shows a significantly reduced bandgap of 1.9 eV compared to its crystalline counterpart (2.82 eV), thus exhibiting much better photocatalytic H₂ production performance under visible light irradiation (Adv. Mater., 2015, 27, 4572-4577).

To our knowledge, there are only several publications reported amorphous metal sulfides as H₂ evolution cocatalysts for other crystalline photocatalysts (Appl. Catal. B, 2018, 232, 446-453; Appl. Catal. B, 2016, 193, 217-225; Appl. Catal. B, 2021, 280, 119455). The function of cocatalyst is similar to electrocatalysts. However, no publications have reported the application of amorphous metal sulfides directly as photocatalysts for photocatalytic H₂ production. In our manuscript, we successfully prepared the amorphous ZnCdS and carefully investigated the photocatalytic activity differences between amorphous ZnCdS and crystalline ZnCdS. According to our studies, the unique photocatalytic performance improvement mechanism comes from the dipole moment effect in the amorphous structure that can provide extra driving forces to promote charge separation, which has not been reported before. We believe that the novelty of our manuscript is high and should have significant impact in relevant fields.

In addition, we also confirmed that amorphous ZnS and CdS also showed enhanced photocatalytic H₂ evolution rates compared to their crystalline counterparts. It should be mentioned that the photocatalytic activity is affected by many factors. Although the

amorphous structure also generates defects that may act as charge recombination centers, the induced strong dipole fields can provide extra driving force for charge separation. Because the positive effect is much higher than the negative effect, the photocatalytic H₂ evolution performance of amorphous ZnCdS can be significantly improved.

As we mentioned in the introduction, “severe charge recombination in the bulk and strong redox capacities of the photogenerated electron-hole pairs that may decompose the photocatalyst itself”. If we can significantly improve the separation and transfer properties of the photogenerated electron-hole pairs, the photogenerated electrons and holes can be consumed in the surface photocatalytic reactions, and thus side-reactions between the photogenerated charge carriers and the photocatalyst itself can be eliminated. From our studies, the AZCS can induce strong dipole fields to provide extra driving forces for charge separation. Therefore, this key issue can be effectively addressed.

Revision has been added in the Introduction Section in the revised manuscript (highlighted in Page 3-4), as shown below:

Main Manuscript (Introduction)

Page 3: If the separation and transfer properties of the photogenerated electron-hole pairs can be significantly improved, the photogenerated electrons and holes can be consumed in the surface photocatalytic reactions, and thus side-reactions between the photogenerated charge carriers and the photocatalyst itself can be eliminated.

Page 4: To the best of our knowledge, this finding has not been reported before in metal sulfide photocatalysts.

4. The current DFT calculation results shown in Figure 1 are not convincing. The amorphous structure in Figure 1b looks completely unreasonable with one- and three-coordinated Cd exposed. Thus, the current theoretical conclusions (including the dipole calculations) lack of credibility and reference value.

Reply: Many thanks for your essential comments. We are sorry that because the super

cell we constructed in the previous version of our manuscript is too small and some parameters of the super cell is not very accurate. We apologize for making the confusion. We have constructed the amorphous structural model with a larger size again, and done the calculations of the deformation charge densities and dipole moments. All the corresponding figures have been updated in Fig. 1. We believe the updated theoretical conclusions should be reliable. We attach Fig. R7 (Fig. 1b) here for your information.

Fig. R7. The atom arrangement of AZCS (Fig. 1b).

5. Fig. 1f and 1h are only a conceptual demonstration and there is no actual data to support them. Also, I cannot understand the Fig. 1h on the energy band bending, which lacks definition and looks strange and even misleading.

Reply: Thanks a lot for your constructive comment. Fig.1g (Fig. 1f in the previous version) is a two-dimensional planar structure obtained based on the structure of AZCS. In order to directly show the changes in positive and negative charge centers caused by different structures, we made this figure based on their dipole moments shown in Fig. 1e (Fig. 1g in the previous version).

To discuss Fig. 1 more logically reasonable, we modified the order of Figs. 1e-g. The updated Fig. 1 is shown in Fig. R8 for your information. The energy band bending in Fig. 1h exists in all polar semiconductors. According to the report in *Chem. Rev.*, 2012, 112, 5520-5551, since ZnCdS in this study is an n-type semiconductor, it exhibits the energy band bending in the n-type case. It is worth noting that polar materials with dipole moments bend their energy bands upward to a more obvious degree than non-polar materials. Based on the calculated dipole fields, we made Fig. 1h as a schematic illustration to help readers understand the larger band bending caused by the dipole

fields can provide extra driving force to promote charge separation. This kind of schematic is very common in many publications (*Nat. Commun.*, 2023, 14, 5742; *Nat. Commun.*, 2020, 11, 2129; *Adv. Funct. Mater.*, 2020, 30, 1908168). We hope that it can be acceptably used in our manuscript.

Fig. R8. Theoretical analysis of dipole field in crystalline and amorphous structures. The atom arrangement and distribution of **a** CZCS and **b** AZCS. Deformation charge densities of **c** CZCS and **d** AZCS on the (011) plane. **e** The calculated dipole moments of AZCS and CZCS along three different crystallographic directions. Schematics of **f** CZCS and **g** AZCS structures with positive and negative charge centers. **h** Schematic of the promotion effect of dipole field on charge transfer. (Fig. 1)

To avoid any misunderstanding, we have added more explanation in the discussion of the revised manuscript, as highlighted in Page 7-8.

Changes to the revised manuscript are shown below.

Main Manuscript (Results)

Page 7-8: According to the calculation results shown in Fig. 1e, the distributions of the positive and negative charge centers of CZCS and AZCS in the y - z plane are demonstrated (Fig. 1f, g). Since the distribution of charge density in CZCS is symmetrical (Fig. 1c), the positive and negative charge centers are close, thus generating a relatively small dipole moment (Fig. 1f). In comparison, the AZCS counterpart is completely asymmetrical (Fig. 1d), and the positive and negative charge centers are separated, thus forming a much stronger dipole moment of 197.01 eÅ along

the (001) direction (Fig. 1g).

To illustrate the contribution of dipole moments to charge separation, a schematic (Fig. 1h) of the energy band structures of CZCS and AZCS during the photocatalytic process were constructed. In the case of CZCS without obvious dipole moments, the energy band bending is too small to drive the directional separation of the photogenerated electrons and holes, and severe charge recombination occurs. However, when the order of all well-arranged unit cells in the ZnCdS crystal are disrupted, it will cause uneven charge distribution in space, thereby generating dipole moments that induce strong dipole fields in the entire photocatalyst. The strong directional dipole field will cause a large energy band bending in AZCS to promote the separation of photogenerated electrons and holes, which can significantly enhance the photocatalytic activity and stability.

6. From the numerical data in Fig. 1g, the dipole direction is completely opposite to the surface exposure (Z) direction, which in principle would change the relative distribution of photoexcited holes and electrons and greatly affect the H₂ evolution efficiency and even change the trend. This is not consistent with the only fivefold improvement in activity.

Reply: Many thanks for your comments. The dipole moment is greatly affected by the crystal structure (*Adv. Energy Mater.*, 2022, 12, 2201208). The dipole moment can be regarded as a built-in electric field. When a sample has a dipole moment, it forms an electric field within itself (*Nat. Commun.*, 2020, 11, 2129; *Angew. Chem. Int. Ed.*, 2023, e202318224). A positive dipole moment creates an electric field in the direction from negative charges to positive charges, while a negative dipole moment creates an electric field in the direction from positive charges to negative charges. The absolute value of the dipole moment decides how strong the driving force can be formed to promote the separation of the photogenerated electrons and holes.

The direction of the dipole moment only affects the spatial separation direction of the photogenerated electrons and holes. For example, a positive dipole moment at the z direction can drive the photogenerated holes (positive charged) moving to the +z direction and photogenerated electrons (negative charged) moving to the -z direction.

In comparison, a negative dipole moment at the z direction can drive the photogenerated electrons (negative charged) moving to the $+z$ direction and photogenerated holes (positive charged) moving to the $-z$ direction. Therefore, both positive and negative dipole moments can lead to the separation of the photogenerated electrons and holes. No matter which direction the electrons and holes are separated from, the photogenerated electrons and holes can be transported to the surface of the photocatalyst for redox reactions.

To avoid any misunderstanding, more explanations have been added in the main manuscript, as highlighted in Page 7. We attach the revision below for your information.

Main Manuscript (Results)

Page 7: When a pair of opposite charges “ $+q$ ” and “ $-q$ ” are separated by a distance “ d ”, an electric dipole is established. The size of dipole is measured by its dipole moment, which is equal to d multiplied by q . The direction of the dipole moment in space is from the negative charge “ $-q$ ” to the positive one “ $+q$ ”^{22,23}. The larger of the absolute value of the dipole moment means the stronger of the extra driving force can be generated in a photocatalyst to promote charge separation.

References

22. Zhang, Y. et al. Visualizing coherent intermolecular dipole-dipole coupling in real space. *Nature* **531**, 623–627 (2016).
23. Li, Z. et al. Dipole field in nitrogen-enriched carbon nitride with external forces to boost the artificial photosynthesis of hydrogen peroxide. *Nat. Commun.* **14**, 5742 (2023).

Reviewer #3 (Remarks to the Author):

This paper introduces a novel amorphous ZnCdS (AZCS) photocatalyst that exhibits remarkable photocatalytic hydrogen evolution performance. The authors attribute this to the asymmetric atom arrangement in AZCS, which induces a strong dipole field that enhances charge separation and transfer. Density functional theory (DFT) calculations are used to elucidate the distinct electronic structure features of crystalline ZnCdS (CZCS) and AZCS materials. However, this paper has some critical problems in both calculation and experimentation that need to be addressed. Only after these issues are

properly resolved, can this work be considered for publication on Nature Communications.

Reply: Thank you very much for your positive feedbacks to further improve the quality of our manuscript. We have carefully done the revision based on your comments.

1. The author should clarify the nature of the AZCS synthesized in their experiment. Is it truly amorphous or rather polycrystalline? In Fig. 2c, it shows some ordered regions in AZCS with a size of several nanometers. Moreover, the electron diffraction patterns and the XRD results in Fig. 2o suggest that the sample is more likely to be polycrystalline with small particle size rather than amorphous. The broadening of the two main peaks in the XRD spectrum could be explained by the Scherrer's formula for polycrystalline particles. On the other hand, amorphous samples usually have only one large broad peak in the XRD spectrum. Therefore, the author may need to reevaluate the structural model of the material.

Reply: Thank you very much for your insightful comments. We are sorry that we might improperly disperse AZCS in ethanol by ultrasonication for HRTEM test in our manuscript. The energy given by ultrasonication may cause a small amount of AZCS to be crystalline. For example, in the case of ultrasonic crystallization, the preparation temperature of traditional long afterglow luminescent materials is at least 1000 °C, while an ultrasonic crystallization method reported in *Adv. Funct. Mater.*, 2019, 29, 1902503 successfully constructed the crystalline material at room temperature. It can be seen that ultrasound can indeed promote crystallization.

To avoid the effect of ultrasonication on the crystallinity of our materials, we dispersed AZCS and CZCS in ethanol without ultrasonication treatment. The diffraction patterns of AZCS and CZCS obtained are shown in Fig. R9 (Figs. 2c, j). It can be clearly observed that AZCS shows the amorphous features without any obvious lattice fringes, and the selected area electron diffraction (SAED) pattern exhibits diffused continuous and thick halo rings without any distinguishable diffraction spots, which is the amorphous features. In comparison, very regular lattice fringes can be observed in CZCS and the SAED pattern shows very clear matrix spots, suggesting the single crystalline features.

Fig. R9. HRTEM and SEAD pattern (inset in) of **a** AZCS and **b** CZCS (Figs. 2c, j)

In an ideal case, amorphous samples usually have only one large broad peak in the XRD pattern. However, in many publications, the amorphous materials exhibit more than one broad peak in the XRD patterns. In the XRD patterns shown in Fig. R10 (part of Fig. 2o), we can observe two broad peaks for AZCS. According to the XRD peaks, we calculated the crystallinity of the samples. As listed in Supplementary Table S1, the crystallinities of AZCS and CZCS are 16.20% and 90.52%, respectively. Therefore, the crystallinity of AZCS is relatively low. We believe that the broad peaks in the XRD pattern should be attributed to the low crystallinity rather than the polycrystalline features. If the crystallinity of a polycrystalline material is very high, the corresponding XRD pattern should show very sharp peaks. We think that the question is how low crystallinity of a material can be called “amorphous material”? It is well accepted that in an amorphous solid there is no long-range order so there are no well-defined scattering planes and therefore no sharp peaks. We think that the XRD pattern of AZCS matches well to amorphous materials.

Fig. R10. XRD pattern of AZCS (part of Fig. 2o)

After carefully researching the literature, the XRD pattern of an amorphous nickel-iron-based electrocatalyst exhibited more than one broad peak (Fig. R11, *Adv. Mater.*, 2019, 31, 1900883). Another work published in *Nano Energy*, 2013, 2, 116-123 reported amorphous NaTaO_x sample that has two broad peaks in the XRD pattern (Fig. R12). Amorphous β-Li₃PS₄ reported in *Adv. Energy Mater.*, 2021, 11, 2101111 exhibit two broad peaks in the XRD pattern (Fig. R13). Amorphous SnO₂ published in *Adv. Mater.*, 2023, 35, 2305587 shows two broad peaks in the XRD pattern (Fig. R14). Amorphous ZnO reported in *Angew. Chem. Int. Ed.*, 2017, 56, 9851-9855 also shows two broad peaks in the XRD pattern (Fig. R15). The XRD pattern we obtained for AZCS shown in Fig. R10 is very similar to the literature (Fig. R11-15). Based on the research of the relevant literature, and the experimental results (HRTEM, SAED pattern, XRD, and Raman spectra), we think it is reasonable to state that the AZCS sample we obtained is the amorphous ZnCdS material. We thank you again for your valuable comments to improve the quality of our manuscript.

Fig. R11. The XRD patterns of crystalline LN and top-down constructed a-LNF(t-d) amorphous samples (Figure 1a in *Adv. Mater.*, 2019, 31, 1900883)

Fig. R12. XRD patterns of amorphous NaTaOx sample prepared at 70 °C for 24 h (denoted as 70°C@24h), followed by calcination at 600 °C for 3 h to form crystalline NaTaO₃ (denoted as Calcined 600°C), sodium tantalum oxide prepared via hydrothermal route at 180 °C 15 h (denoted as 180°C@15h) and 115 h (denoted as 180°C@115h). (Figure 1e in *Nano Energy*, 2013, 2, 116-123)

Fig. R13. XRD patterns of the amorphous (denoted as A), low crystallinity (denoted as LC), middle crystallinity (denoted as MC), high crystallinity (denoted as HC) β-Li₃PS₄ samples. (Figure 4 in *Adv. Energy Mater.*, 2021, 11, 2101111)

Fig. R14. XRD pattern of amorphous SnO₂. (Figure S8b in Adv. Mater., 2023, 35, 2305587)

Fig. R15. XRD patterns of amorphous ZnO (a-ZnO NCs) and crystalline ZnO (c-ZnO NCs). (Figure S6 in Angew. Chem. Int. Ed., 2017, 56, 9851-9855)

Changes to the revised manuscript are shown below.

Main Manuscript (Results)

Page 9: The HRTEM image of AZCS shows disordered atomic arrangement without obvious lattice fringes (Fig. 2c), and the selected area electron diffraction (SAED) pattern exhibits diffused continuous and thick halo rings without any distinguishable diffraction spots (inset in Fig. 2c), indicating the amorphous feature of AZCS.

2. The authors claim that the amorphous structure induces strong dipole fields that promote charge separation. However, this argument is not convincing for two reasons. First, the structural model of the amorphous ZnCdS is not well supported by experimental evidence. Second, the dipole changes are calculated from small super cell model contains only two periodic layers, which is too simplistic to capture the complex disorder and local variations in the amorphous structure.

Reply: Thank you very much for your excellent comments. We are sorry that there may be some misunderstandings in our structural model of amorphous ZnCdS. In our response to your 1st comment, we have confirmed that the AZCS we obtained is amorphous ZnCdS.

We first constructed the model structure of CZCS to obtain the AZCS counterpart. We used the XRD pattern obtained experimentally and analyzed it with Jade to obtain the unit cell information of CZCS (JCPDS-ICDD:97-060-0508), and then constructed the structure of CZCS as shown in Fig. R16a based on the unit cell information. Since the amorphous structures don't have a lattice model, it is difficult to verify the specific structure experimentally (ACS Nano, 2021, 15, 739-750; Adv. Sci., 2022, 9, 2201903; Angew. Chem. Int. Ed., 2017, 56, 9851-9855; J. Am. Chem. Soc., 2019, 141, 5856-5862). Here we model AZCS (Fig. R16b) through a quenching method according to the literature (Nat. Commun., 2022, 13, 7205; Angew. Chem. Int. Ed., 2023, 62, e202216658). More specifically, we applied molecular dynamics simulation for the CZCS structure to be heated at 3000 K to remove the memory effect from the initial sites. Secondly, the models were cooled down to 1400 K and relaxed at this temperature for a stable liquid. Then, we rapidly reduced the temperature to 300 K with a cooling rate of 33.3 K/ps, and finally, the models were maintained at 300 K for 30 ps to collect the trajectories of the atoms. We think this newly built model is well supported by the experimental

evidence.

Fig. R16 The model structure of **a** CZCS and **b** AZCS. (**Fig. 1a, b**)

Sorry for ignoring size of the super cell we used for calculations in the experimental details. We have reconstructed a large super cell model to obtain more reliable results based on your excellent comments. The dipole changes were calculated from a $4 \times 4 \times 3$ supercell model (containing 192 atoms) with a cell parameter $> 16 \text{ \AA}$, which can eliminate the effect of periodicity on the photocatalytic properties. The amorphous ZnCdS structure is also at the same size, which should be large enough for the calculation of the dipole moments. All the calculations were conducted based on the newly built supercell model, and the corresponding discussion in the main text has been updated. If we use a larger supercell model for calculation, because the system contains much more atoms, the calculations would be much slower and it would take much more time to get the results. Considering that we only have 4 weeks for revision, we don't have enough time to build an even larger supercell model. We hope that our newly-built $4 \times 4 \times 3$ supercell model (containing 192 atoms) is acceptable for DFT calculations.

According to a previous study (*Energy Procedia*, 2012, 29, 291-299) for DFT study of structural and electronic properties of amorphous TiO_2 , three model samples of bulk amorphous TiO_2 with super cell dimensions of $2 \times 2 \times 3$ (72 atoms), $2 \times 2 \times 4$ (96 atoms), and $3 \times 3 \times 4$ (216 atoms) were prepared by molecular dynamics simulations. The results showed that their calculated electronic properties are very similar. Therefore, we think the super cell model with dimensions of $4 \times 4 \times 3$ (containing 192 atoms) applied for calculations should be large enough to get reliable results.

Changes to the revised manuscript are shown below.

Main Manuscript (Results)

Page 5-6: In the hexagonal system of crystalline ZnCdS (CZCS), each Zn or Cd atom is connected to four S atoms with a perfect layered structure (Fig. 1a and top view in Supplementary Fig. S1a), while AZCS exhibits a random arrangement of the ZnS₄ and CdS₄ tetrahedrons (Fig. 1b and top view in Supplementary Fig. S1b). Supplementary Fig. S2 exhibits the DFT energy as a function of time at 300 K for CZCS and AZCS, respectively. CZCS possesses a lower energy than its AZCS counterpart for the stabilization order, demonstrating the higher structural stability. Moreover, the deformation charge density distributions of CZCS and AZCS along the (011) plane are shown in Figs. 1c, d, respectively. The charge distribution of CZCS is very uniform and highly order, while AZCS demonstrates the random distribution of the deformation charge density, which is attributed to the different atomic arrangement and distribution in CZCS and AZCS.

Main Manuscript (Methods-DFT calculations)

Page 29-30: Crystalline ZnCdS was a hexagonal crystal system, and the unit cell model was obtained based on the XRD results. The initial supercell contained 192 atoms of ZnCdS (4 × 4 × 3) with a cell parameter >16 Å. The amorphous models were obtained by using the melt-quenched process: Firstly, the primitive models were fully melted at 3000 K to remove the memory effect from the initial sites. Secondly, the models were cooled down to 1400 K and relaxed at this temperature for a stable liquid. Then, we rapidly reduced the temperature to 300 K with a cooling rate of 33.3 K/ps, and finally, the models were maintained at 300 K for 30 ps to collect the trajectories of the atoms.

3. In the manuscript, the authors did not explain how the structural models of CZCS and AZCS were constructed for the DFT calculation. In particular, the details of the AZCS model is unclear. According to Fig. 1, the AZCS model seems to be derived from the CZCS crystal model by introducing local distortions, but still retains the alternating layered structure of ZnS and CdS. This contradicts the “random arrangement” of [ZnS₄] and [CdS₄] units that the author claimed in the main text (line 88, page 5).

Reply: We appreciate your valuable comments. We are sorry for ignoring the important information of how the structural models of CZCS and AZCS were constructed for DFT

calculations. We used the unit cell information of CZCS to construct the structural model and quenched the CZCS and applied molecular dynamics simulation to construct the structural model of AZCS (please refer to our response to your Comment 2nd for more information). More details have been highlighted in the DFT calculations Section in the revised manuscript.

We are sorry that the description of “random arrangement of [ZnS₄] and [CdS₄] units” in the main text is not well supported by the constructed model in the previous version of our manuscript. Based on your Comment 2nd, we have constructed a new supercell model with dimensions of 4 × 4 × 3 (containing 192 atoms) for DFT calculations. In our newly constructed model, it can be seen from the top view of CZCS and AZCS (Fig. R17) that the structure of AZCS is random arrangement of [ZnS₄] and [CdS₄] units.

Fig. R17 The top view of the atom arrangement of a CZCS and b AZCS.

(Supplementary Fig. S1)

Changes to the revised manuscript are shown below.

Main Manuscript (Methods-DFT calculations)

Page 29-30: All calculations were performed using the density functional theory (DFT), as implemented in the Vienna ab initio simulation package^{60,61}. The projector augmented-wave (PAW) method and Perdew-Burke-Ernzerhof generalized gradient approximation (GGA-PBE) were used for the exchange correlation functionals^{62,63}. The time step was set to 3 fs and only the Γ point was sampled from the Brillouin zone. The canonical (NVT) ensemble with the Nose–Hoover thermostat was applied to control the temperature and the pressure in AIMD simulations. Crystalline ZnCdS was a hexagonal crystal system, and the unit cell model was obtained based on the XRD results. The initial supercell contained 192 atoms of ZnCdS (4 × 4 × 3) with a cell parameter >16 Å. The amorphous models were obtained by using the melt-quenched

process: Firstly, the primitive models were fully melted at 3000 K to remove the memory effect from the initial sites. Secondly, the models were cooled down to 1400 K and relaxed at this temperature for a stable liquid. Then, we rapidly reduced the temperature to 300 K with a cooling rate of 33.3 K/ps, and finally, the models were maintained at 300 K for 30 ps to collect the trajectories of the atoms. For the calculations of electronic structure, the energy cutoff of the PAW basis was set to 450 eV with a force convergence of 0.02 eV and a $2 \times 2 \times 1$ k-points grid was selected for the Brillouin zone sampling.

4. The DFT calculated energy of AZCS and CZCS is not reported. The authors need to compare the stability of the theoretical models AZCS and CZCS and verify them with their experimental results.

Reply: Many thanks for your valuable suggestions. We confirmed that the DFT calculated energy of AZCS and CZCS are about -513 and -521 eV at 300 K (Fig. R18), respectively. The higher Gibbs energy of a material means the lower stability of the materials. According to the photocatalytic H₂ evolution performances of AZCS and CZCS without any cocatalyst, the photocatalytic activity of AZCS decreases gradually with the irradiation of time (Fig. R19). However, the photocatalytic activity of CZCS is relatively stable with the irradiation of time. Therefore, AZCS is less stable than CZCS, which is consistent to the calculation results.

Fig. R18 DFT energy of CZCS and AZCS as a function of time at 300 K.

(Supplementary Fig. S2)

Fig. R19. Photocatalytic H₂ evolution of AZCS and CZCS loaded without cocatalyst. (part of **Supplementary Fig. S10**)

Changes to the revised manuscript are shown below.

Main Manuscript (Results)

Page 5-6: Supplementary Fig. S2 exhibits the DFT energy as a function of time at 300 K for CZCS and AZCS, respectively. CZCS possesses a lower energy than its AZCS counterpart for the stabilization order, demonstrating the higher structural stability.

Supplementary Information

Page 8-9, Supplementary Discussion

Owing to the sluggish photocatalytic reaction without a cocatalyst, the difference between the photocatalytic performance of AZCS and CZCS gradually decreases. It should be mentioned that the photocatalytic H₂ production rate of AZCS gradually decreases with the irradiation time, while its CZCS counterpart exhibits a relatively stable photocatalytic H₂ production rate during the same irradiation time, indicating that AZCS without cocatalyst is less stable than CZCS. These experimental results are consistent with the DFT calculations shown in Supplementary Fig. S2.

5. The separation of photo generated electrons and holes in AZCS may be validated by the spatial distribution of the HOMO and LUMO orbitals.

Reply: Thank you for your excellent suggestions. We have calculated the spatial distribution of the HOMO and LUMO orbitals of AZCS and CZCS. As shown in Fig. R20 (Supplementary Fig. S3), the molecular bandgap of CZCS shows 2.28 eV, while AZCS shows 0.41 eV. The spatial distribution of the HOMO and LUMO orbitals of AZCS is 1.87 eV smaller than that of CZCS. This trend is consistent to the experimental measurements (Fig. 4a-e). It should be mentioned that the calculated bandgap of a semiconductor is usually lower than the experimentally measured bandgap. The calculated bandgap cannot be compared with the experimental bandgap. However, the change trend of the calculated bandgap of the AZCS and CZCS can be reasonably compared.

Fig. R20 HOMO and LUMO distribution of **a** CZCS and **b** AZCS. (Supplementary Fig. S3)

Changes to the revised manuscript are shown below.

Main Manuscript (Results)

Page 8: To demonstrate the effect of atomic arrangement and distribution on the bandgap, the HOMO and LUMO orbitals of CZCS and AZCS were also calculated. As shown in Supplementary Fig. S3, the HOMO charge densities are strongly localized at the S atoms, and the LUMO charge densities are strongly localized at the Zn, Cd, and

S atoms, which is consistent to the literature that the valence band (VB) maximum of ZnCdS is mainly dominated by the 3p orbital of the S atom, while the conduction band (CB) minimum of ZnCdS is mainly contributed by the hybridization of the 4s orbital of the Zn atom, the 5s orbital of the Cd atom and the 3p orbital of the S atom²⁴. In addition, the bandgap of AZCS is narrower than its CZCS counterpart, suggesting that AZCS can absorb a broader range of light.

Page 20: The reduced bandgap of AZCS is consistent to the DFT calculation results shown in Supplementary Fig. S3.

Reference

24. Wang, W. *et al.* Band-Gap Modulation for Enhancing NO Photocatalytic Oxidation over Hollow ZnCdS: A Combined Experimental and Theoretical Investigation. *J. Phys. Chem. C* 126, 3967–3979 (2022).

6. In the Methods section, it lacks some key information in the calculation details, such as the size of the super cell and the construction method of the AZCS structural model. These should be clearly stated and justified.

Reply: Thanks for your suggestions. We are sorry for missing the key information in the calculation details. All the required information has been added in the revised manuscript, as highlighted in Page 29-30.

The added information is shown below for your information:

Main Manuscript (Methods – DFT calculations)

Page 29-30: All calculations were performed using the density functional theory (DFT), as implemented in the Vienna ab initio simulation package^{60,61}. The projector augmented-wave (PAW) method and Perdew-Burke-Ernzerhof generalized gradient approximation (GGA-PBE) were used for the exchange correlation functionals^{62,63}. The time step was set to 3 fs and only the Γ point was sampled from the Brillouin zone. The canonical (NVT) ensemble with the Nose–Hoover thermostat was applied to control the temperature and the pressure in AIMD simulations. Crystalline ZnCdS was a hexagonal crystal system, and the unit cell model was obtained based on the XRD results. The initial supercell contained 192 atoms of ZnCdS ($4 \times 4 \times 3$) with a cell

parameter $>16 \text{ \AA}$. The amorphous models were obtained by using the melt-quenched process: Firstly, the primitive models were fully melted at 3000 K to remove the memory effect from the initial sites. Secondly, the models were cooled down to 1400 K and relaxed at this temperature for a stable liquid. Then, we rapidly reduced the temperature to 300 K with a cooling rate of 33.3 K/ps, and finally, the models were maintained at 300 K for 30 ps to collect the trajectories of the atoms. For the calculations of electronic structure, the energy cutoff of the PAW basis was set to 450 eV with a force convergence of 0.02 eV and a $2 \times 2 \times 1$ k-points grid was selected for the Brillouin zone sampling.

7. The U values used in the calculations is important and could lead to inaccurate results. The authors should use Hubbard U and exchange parameter J or Ueff to represent the U values, and provide a detailed procedure of how they were tested and chosen.

Reply: We appreciate your valuable comments. We are sorry for missing the details of choosing the U values for Cd and Zn. The details are shown below:

The requisite exchange-correlation (XC) potentials for structural and elastic properties have been computed with the Perdew–Burke–Ernzerhof generalized gradient approximation (PBE-GGA) scheme. On the other hand, we have employed the modified Becke–Johnson (mBJ)-GGA and GGA+U schemes to compute XC potentials for electronic properties. In the GGA+U approach, the PBE-GGA based electronic properties have been calculated as an introductory step. Subsequently, Coulomb interactions between the localized 3d and 4d electrons in the Zn and Cd atoms, respectively, have been incorporated with Coulomb self-interaction potentials through the Hubbard parameter U. As shown in Fig. R21, when the U value of Zn is higher than 7 and the U value of Cd is higher than 4, the bandgap of ZnCdS is stable. Therefore, U values have been utilized as 7.0 and 4.0 eV for Zn and Cd atoms for the Coulomb corrections to the Zn 3d and Cd 4d states, respectively.

Fig. R21. Bandgaps obtained by adding different values of U to **a** Zn and **b** Cd.

Changes to the revised manuscript are shown below.

Main Manuscript (Methods – DFT calculations)

Page 30: The U_{eff} values of Zn and Cd were calculated based on the bandgap of ZnCdS. When the U_{eff} values of Zn and Cd were higher than 7.0 and 4.0 eV, the bandgap ZnCdS trends to be stable. Therefore, the Coulomb interaction U_{eff} values were set to 7.0 and 4.0 eV to describe the 3d electrons of Zn and 4d electrons of Cd, respectively, which are also consistent to the literature^{64,65}.

References

64. Biswas, A., Meher, S. R. & Kaushik, D. K. Electronic and Band Structure calculation of Wurtzite CdS Using GGA and GGA+U functionals. *J. Phys. Conf. Ser.* **2267**, 012155 (2022).
65. Jiang, H., Gomez-Abal, R. I., Rinke, P. & Scheffler, M. First-principles modeling of localized d states with the GW@LDA+U approach. *Phys. Rev. B Condens. Matter Mater. Phys.* **82**, 045108 (2010).

REVIEWERS' COMMENTS

Reviewer #1 (Remarks to the Author):

ok

Reviewer #2 (Remarks to the Author):

The authors have made efforts to address the concerns raised, and this manuscript is now worthy of recommendation for publication.

Reviewer #3 (Remarks to the Author):

The authors carefully revised the manuscript and thoughtfully addressed my previous comments. They have redone the DFT calculations and provided additional details and data to support their conclusions. I think the paper has been significantly improved and meets the standards for publication in Nature Communications. I have no further questions or suggestions and recommend its acceptance.

Response to Reviewers' Comments

Reviewer #1 (Remarks to the Author):

ok

Reply: We appreciate your helpful comments to improve the quality of our manuscript.

Reviewer #2 (Remarks to the Author):

The authors have made efforts to address the concerns raised, and this manuscript is now worthy of recommendation for publication.

Reply: We are grateful for your guidance and constructive comments to improve the quality of our work.

Reviewer #3 (Remarks to the Author):

The authors carefully revised the manuscript and thoughtfully addressed my previous comments. They have redone the DFT calculations and provided additional details and data to support their conclusions. I think the paper has been significantly improved and meets the standards for publication in Nature Communications. I have no further questions or suggestions and recommend its acceptance.

Reply: Thank you very much for all your valuable suggestions to help improving the quality of our work, and many thanks for your approval on our revision.